# Topological Signatures of Adversaries in Multimodal Alignments

**Minh Vu** [1]  **Geigh Zollicoffer** [1]  **Huy Mai** [2]  **Ben Nebgen** [1]  **Boian Alexandrov** [1]  **Manish Bhattarai** [1]

## Abstract

Multimodal Machine Learning systems, particularly those aligning text and image data like CLIP/BLIP models, have become increasingly prevalent, yet remain susceptible to adversarial attacks. While substantial research has addressed adversarial robustness in unimodal contexts, defense strategies for multimodal systems are underexplored. This work investigates the topological signatures that arise between image and text embeddings and shows how adversarial attacks disrupt their alignment, introducing distinctive signatures. We specifically leverage persistent homology and introduce two novel Topological-Contrastive losses based on Total Persistence and Multi-scale kernel methods to analyze the topological signatures introduced by adversarial perturbations. We observe a pattern of monotonic changes in the proposed topological losses emerging in a wide range of attacks on image-text alignments, as more adversarial samples are introduced in the data. By designing an algorithm to backpropagate these signatures to input samples, we are able to integrate these signatures into Maximum Mean Discrepancy tests, creating a novel class of tests that leverage topological signatures for better adversarial detection.

## 1. Introduction

In recent years, the rapid advancement of artificial intelligence (AI) has led to the development of increasingly complex multimodal systems that integrate diverse streams of data, including text, images, audio, and graphs. As these technologies become more pervasive, they also face significant vulnerabilities, particularly from adversarial attacks. Adversarial attacks exploit inherent weaknesses in machine learning (ML) models by introducing subtle perturbations that can maliciously alter the system's outputs (Goodfellow, 2014; Brendel et al., 2017). While substantial research has focused on safeguarding AI models against adversarial attacks in unimodal contexts, the challenges presented by the multimodal landscape remain understudied. Notably, although adversarial attacks in multimodal settings have advanced rapidly (Zhang et al., 2022; Zhou et al., 2023), defense strategies have yet to fully consider the unique characteristics of multimodal systems, which are crucial for enhancing their robustness against such threats.

In this study, we focus on multimodal alignments, typically appearing in CLIP (Radford et al., 2021) and BLIP (Li et al., 2022), where the image and text data embeddings are aligned for downstream predictions. Based on the Manifold Hypothesis (Goodfellow et al., 2016), in many instances—such as natural images—the support of the data distribution stays on a low-dimensional manifold embedded within Euclidean space. Many interesting features and patterns of those data can be captured through the topological properties of that manifold as well as the manifold of their neural network's embeddings. Particularly, through extensive experiments based on persistent homology, an emerging technique in topological data analysis (TDA) to study those manifolds, we show that adversarial attacks alter the image-text topological alignment and introduce distinctive topological signatures. We further demonstrate that the presence of adversarial examples in an image batch can be more effectively detected with those signatures.

The key results of this work are highlighted in Fig. 1. Using the ImageNet (Deng et al., 2009) and CIFAR-10 (Krizhevsky, 2009) datasets with CLIP-ViT-B/32 and CLIP-ViT-L/14@336px, respectively, we demonstrate our proposed Total Persistence (TP) loss $\mathcal{L}_{TP}^{\alpha}$ and Multi-scale Kernel (MK) loss $\mathcal{L}_{MK}^{\sigma}$ under varying proportions of adversarial samples in the data batch. Intuitively, these losses capture the mismatch in the corresponding topological signatures between the image and text embeddings. The results clearly show a proportional change in these losses as the percentage of adversarial data increases, emphasizing the sensitivity of our topological measures to adversarial perturbations. In the third and fourth columns, we demonstrate how the Maximum Mean Discrepancy (MMD) (Grosse et al., 2017; Gao et al., 2021) tests, based on topological features derived from the TP and MK losses (TPSAMMD and MK-

[1]Theoretical Division, Los ALamos National Laboratory, Los Alamos, NM, USA [2]Independent. Correspondence to: Minh Vu <mvu@lanl.gov>.

*Proceedings of the 42nd International Conference on Machine Learning*, Vancouver, Canada. PMLR 267, 2025. Copyright 2025 by the author(s).

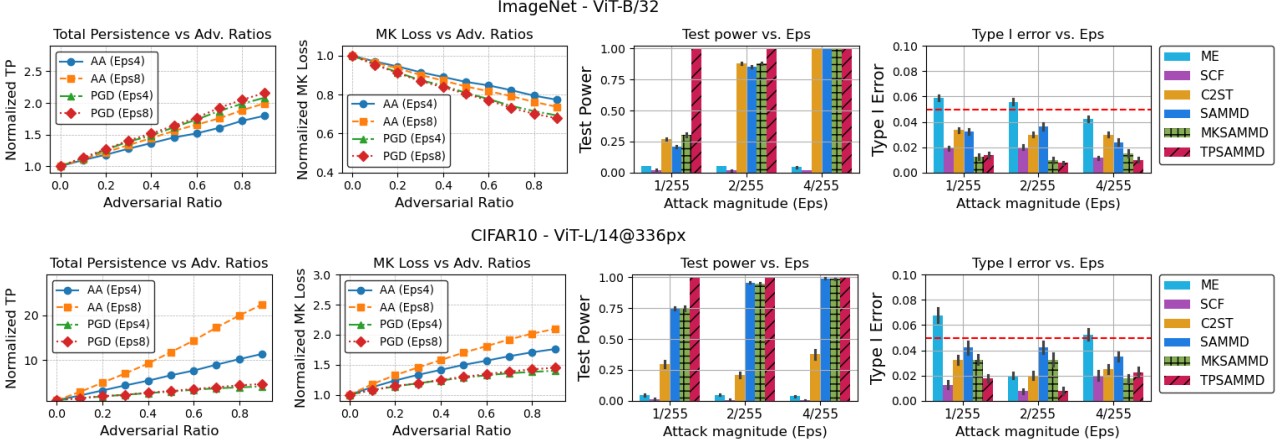

*Figure 1.* Highlights of our results in ImageNet (top) and CIFAR10 (bottom) in CLIP's alignment: (1st and 2nd columns) The TP and MK losses of the data batch monotonically change as the proportion of adversaries (AA for AutoAttack and PGD for Projected Gradient Descent) in the batch increases. (3rd and 4th columns) Utilizing the topological signatures derived from TP and MK losses can significantly improve the Test power of existing MMD solutions, e.g., TPSAMMD and MKSAMMD vs SAMMD, while keeping the Type I errors under control (e.g., 5% error). The results are shown for Autoattack with $L_\infty$.

SAMMD), enhance the test power and control Type-I error in existing MMD solutions, highlighting the effectiveness of our topological approach in detecting multimodal adversarial attacks. This work's main contributions are:

- We introduce two novel Topological-Contrastive (TC) losses, $\mathcal{L}_{TP}^\alpha$ and $\mathcal{L}_{MK}^\sigma$, based on Total Persistence (Divol & Polonik, 2019) and Multi-scale Kernel (Reininghaus et al., 2014), measuring the topological differences between image embeddings and text embeddings in multimodal alignment. We provide extensive experiments showing the presence of adversaries results in a clear distinction in the TC losses, i.e., in most settings, the TC losses monotonically change when more adversarial data is in the data batch. We also provide a theoretical justification for the increase of TP via Poisson Cluster Process modeling.

- We integrate both TC losses into state-of-the-art (SOTA) MMD tests that differentiate between clean and adversarial data, creating a novel class of MMD tests based on TC features. Specifically, we design an algorithm to back-propagate the TC losses to the input samples, resulting in the TC features capturing how much each sample contributes to the global topological distortion in the data batch. These features are then used to enhance the MMD test.

- We conduct extensive experiments in 3 datasets (CIFAR-10, CIFAR-100, and ImageNet), 5 CLIP embeddings (ResNet50, ResNet101, ViT-B/16, ViT-L/14, and ViT-L/14@336px), 3 BLIP embeddings ( ViT-B/14, ViT-B/129, and ViT-B/129-CapFilt-L), and 6

adversarial generation methods (FGSM, PGD, AutoAttack, APGD, BIM, and Carlini-Wagner (CW)) to demonstrate the advantages of the two above findings.

Our paper is organized as follows. Sect. 2 provides the background and related work of this study. Sect. 3 formulates our proposed TC losses and demonstrates their capabilities in monitoring alignment of multimodal adversaries. Sect. 4 shows how we can leverage the TC losses for MMD-based adversarial detection. Sect. 5 reports our experimental results, and Sect. 6 concludes this paper.

## 2. Related Work and Preliminaries

**The Two-sample test and the MMD:** Given samples from two distributions $\mathbb{P}$ and $\mathbb{Q}$, a two-sample test assesses whether to reject the null hypothesis that $\mathbb{P} = \mathbb{Q}$ based on a test statistic that measures the distance between the samples. One such test statistic is the MMD, which quantifies the distance between the embeddings of the probability distributions into a reproducing kernel Hilbert space (RKHS) (Gretton et al., 2012):

$$\text{MMD}(\mathbb{P}, \mathbb{Q}; \mathcal{H}_k) := \sup_{f \in \mathcal{H}, \|f\| \leq 1} |\mathbb{E}[f(X)] - \mathbb{E}[f(Y)]| \quad (1)$$

Here, $k$ is a bounded kernel associated with the RKHS $\mathcal{H}_k$ (i.e., $|k(\cdot, \cdot)| < \infty$), $\|\cdot\| := \|\cdot\|_{\mathcal{H}_k}$ and X and Y are random variables sampled from $\mathbb{P}$ and $\mathbb{Q}$, respectively. Gretton et al. demonstrated that the MMD equals zero *if and only if* $\mathbb{P} = \mathbb{Q}$, indicating that the MMD can be used to determine whether two distributions are identical.

**Adversarial Detection in Unimodal/Multimodal ML:** In

the unimodal context, many defense methods have been proposed to safeguard against adversaries (Costa et al., 2024). For our scope, we focus on statistical methods detecting the presence of adversaries based on MMD. Specifically, the paper (Grosse et al., 2017) introducing a Gaussian kernel-based two-sample test is one of the first works employing the MMD to detect adversaries. Later, Sutherland et al. (Sutherland et al., 2016) and Liu et al. (Liu et al., 2020) developed MMD tests with optimized Gaussian kernels and deep kernels, respectively. Gao et al. (Gao et al., 2021) further proposed applying the MMD test to the semantic features (SAMMD) of the attacked models, which significantly improved test power. In addition to MMD-based methods, numerous other detection techniques leverage recent advancements in diffusion models (Zhang et al., 2023), Bayesian uncertainty estimates (Feinman et al., 2017), manifold learning (Ma et al., 2018; Park et al., 2022), Gaussian discriminant analysis (Lee et al., 2018), unsupervised clustering (Cohen et al., 2020; Mao et al., 2020), and neural network classifiers (Li & Li, 2017).

Adapting adversarial detection techniques towards multimodal systems has recently gained substantial interest (Zhang et al., 2024a). Zhang et al. suggests utilizing a variety of mutations to disturb adversarial perturbations. Hsu et al. empirically identifies the model's weights that contribute to unwanted outputs and safeguards the model by utilizing LORA (Hu et al., 2022). Prompt optimization techniques have also been used to better inform the model of possible attacks (Wang et al., 2024b). There also exists recent evidence that hidden states of multimodal systems contain signatures of adversarial text data and can be utilized for detection (Zhao et al., 2024; Wang et al., 2024a).

**Multimodal Alignment:** CLIP (Radford et al., 2021) and BLIP (Li et al., 2022) are multimodal systems developed by OpenAI and Salesforce respectively that bridge the gap between natural language and visual understanding. Unlike traditional systems that are trained for specific tasks, they are trained to understand a wide variety of visual concepts in a zero-shot manner by leveraging large-scale image-text pairs. Both systems consist of an image encoder, typically a convolutional neural network (e.g., ResNet) or a Vision Transformer (ViT) that processes input images and maps them to a high-dimensional embedding space, and a text encoder, usually a Transformer (e.g., GPT-like architecture) that processes input text descriptions and maps them to the same embedding space as the image encoder. Those image and text embeddings are then aligned to perform a variety of tasks without additional computation. For instance, CLIP and BLIP can conduct zero-shot image classification by comparing the image embeddings to class labels' embeddings. Thus, adversaries can be considered as specially crafted inputs designed to disrupt the image-text alignment.

**Topological data analysis (TDA):** is an emerging field in mathematics that applies the concepts of topology into practical, real-world applications. It has significant uses in areas such as data science, robotics, and neuroscience. TDA employs advanced tools from algebraic topology to analyze the inherent topological structures in data, uncovering insights that traditional metric-based methods may overlook. The most widely used technique in modern TDA is persistent homology, developed in the early 2000s by Gunnar Carlsson and his collaborators. We recommend consulting (Ghrist, 2014; Edelsbrunner & Harer, 2022) for a comprehensive overview of persistent homology and TDA.

Given a point cloud $X$, this work considers the Vietoris–Rips complex (VR) (Vietoris, 1927) of $X$ at scale $\epsilon$ (where $0 \leq \epsilon < \infty$), denoted by $\mathfrak{R}_\epsilon(X)$. This complex includes all simplices (i.e., subsets) of $X$ such that every pair of points within a simplex satisfies $\text{dist}(x_i, x_j) \leq \epsilon$ for all $x_i, x_j \in X$. Since the VR complexes satisfy the relation $\mathfrak{R}_\epsilon(X) \subseteq \mathfrak{R}_{\epsilon'}(X)$ for $\epsilon \leq \epsilon'$, they constitute a *filtration* and can track the evolution of the homology groups (i.e., topological invariants) as $\epsilon$ increases (Edelsbrunner & Harer, 2022). Each topological feature is monitored by its *birth time* $\epsilon = b$ when it appears and its *death time* $\epsilon = d$ when it disappears. They are the central concepts to understand the persistence of topological signatures within the data. For a given dimension $i$, the birth-death pairs $(b, d) \in \mathbb{R}^2_{\geq 0}, (b < d)$ can be encoded in a persistence diagram $D_i(X)$ corresponding to the features in the $i$-th homology group. Intuitively, the difference between death and birth indicates the significance of a feature. Readers can refer to Fig. 2 for illustrations of the VR filtration and the persistence diagrams.

## 3. Topological Signatures of Adversaries against Multimodal Alignment

Our analysis of multimodal adversaries is based on the intuition that the adversarial perturbations in one data stream can potentially alter or destroy the alignment between that data and another data stream. The problem is in how to capture this behavior, analyze, and leverage it for better adversarial detection. Measuring multimodal alignment is not trivial because the data streams not only come from different domains, have different representations, and semantic meanings, but also lack index alignment. We tackle that question through the lens of TDA: we first capture the topological structures from the point clouds of representations of each data stream. We then compare the extracted topological features to measure the alignment. In particular, we rely on the *total persistence* (Divol & Polonik, 2019; Edelsbrunner & Harer, 2022) and the *Multi-scale Kernel* (Reininghaus et al., 2014), two concepts of persistence homology, to quantify topological information.

This section first describes our proposed topological con-

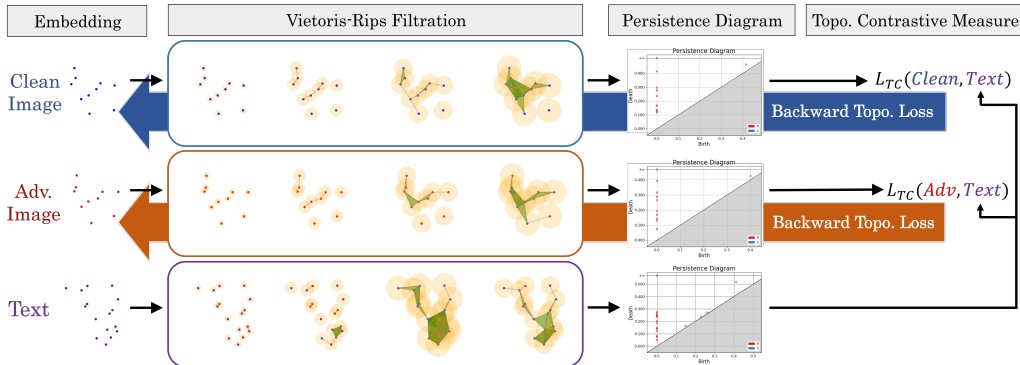

*Figure 2.* The computations of the TC loss $\mathcal{L}_{TC} \in \{\mathcal{L}_{TP}^\alpha, \mathcal{L}_{MK}^\sigma\}$ and the backward computations of the topological features.

trastive losses, i.e., the TP and the MK losses (Subsect. 3.1). Then, results from extensive experiments on CLIP and BLIP models will show that the presence of adversaries in the input data batch introduces distinct topological signatures captured by TP and ML losses (Subsect. 3.2). We further attempt to explain the observed monotonic increasing behavior of TP by modeling the logits as a Poisson Cluster Process (PCP) (Subsect. 3.3). With the assumption that the adversaries' logits are less tightly clustered to the predicted classes, PCP modeling enables Monte Carlo simulation to compute TP and demonstrate why adversarial logits generally result in higher 0-th order TP, providing insights into how adversarial attacks affect the logits' topology.

### 3.1. Topological Contrastive Losses

This study focuses on two homology signatures commonly employed in TDA: while TP measures the cumulative importance of topological features across all dimensions, serving as a holistic summary of a dataset's topology, the MK facilitates comparisons between datasets by analyzing their topological features at multiple scales. We choose these two homologies for this study for their distinctive behaviors with adversaries (Subsect.3.2) and their capability to be back-propagated for adversarial detection (Sect.4).

**TP Loss**: For a given dimension $i$, the $\alpha$-total persistence of dimension $i$ is computed on the persistence diagram $D_i(X)$ (Divol & Polonik, 2019):

$$\mathrm{Pers}_i^\alpha(X) := \sum_{(b,d) \in D_i(X)} (d-b)^\alpha \qquad (2)$$

The TP loss of order $\alpha$ between two point clouds is the summation of the difference at all homology groups:

$$\mathcal{L}_{TP}^\alpha(X,Y) = \sum_i |\mathrm{Pers}_i^\alpha(X) - \mathrm{Pers}_i^\alpha(Y)| \qquad (3)$$

The TP (Eq.2) can be considered as a fundamental quantity for persistence homology analysis as it is closely related to the *Wasserstein* distance between point clouds. We would

refer readers to (Divol & Polonik, 2019) for more details on how the loss can be used to capture topological information.

**MK Loss:** The loss is formulated based on the Multi-scale kernel $k_\sigma : \mathcal{D} \times \mathcal{D} \to \mathbb{R}$ introduced by Reininghaus et al., acting on persistence diagrams of point clouds $X$ and $Y$:

$$k_\sigma(D_i(X), D_i(Y)) :=$$
$$\frac{1}{8\pi\sigma} \sum_{p \in D_i(X), q \in D_i(Y)} e^{-\frac{\|p-q\|_2^2}{8\sigma}} - e^{-\frac{\|p-\bar{q}\|_2^2}{8\sigma}} \qquad (4)$$

where $p$ and $q$ are the birth-death pairs from the corresponding persistence diagrams, and $\bar{q} = (d, b)$ denotes the mirror of $q = (b, d)$ through the diagonal. Notably, the MK is proved to be 1-*Wasserstein* stable (Theorem 2. Reininghaus et al.), and is a robust summary representation of data's topological features. For our purpose, we define the MK loss of scale $\sigma$ between two point clouds by:

$$\mathcal{L}_{MK}^\sigma(X,Y) = \sum_i k_\sigma(D_i(X), D_i(Y)) \qquad (5)$$

**Topological Contrastive Losses for Multimodal Alignment:** Fig. 2 describes how we use our TC losses $\mathcal{L}_{TC} \in \{\mathcal{L}_{TP}^\alpha, \mathcal{L}_{MK}^\sigma\}$ to analyze multimodal alignment. The forward left-to-right arrows indicate the computation of TP and MK losses. In CLIP and BLIP, the images and text representations are aligned in a shared embedding space. We first extract the those logit/embeddings before the alignment step, treating these embeddings as point clouds. Next, the corresponding VR filtrations $\{\mathfrak{R}_\epsilon\}_\epsilon$ are constructed, followed by the generation of persistence diagrams $\{D_i\}_i$. The TC losses are then computed using Eq. 3 for the TP and Eq. 5 for the MK losses, with $X$ and $Y$ representing the image and text embeddings, respectively.

### 3.2. Monotonic Behaviors of Adversarial TC losses

We now present a key finding of this work. Through extensive experiments utilizing the proposed TC losses conducted across models, adversarial attacks, and datasets, we observe

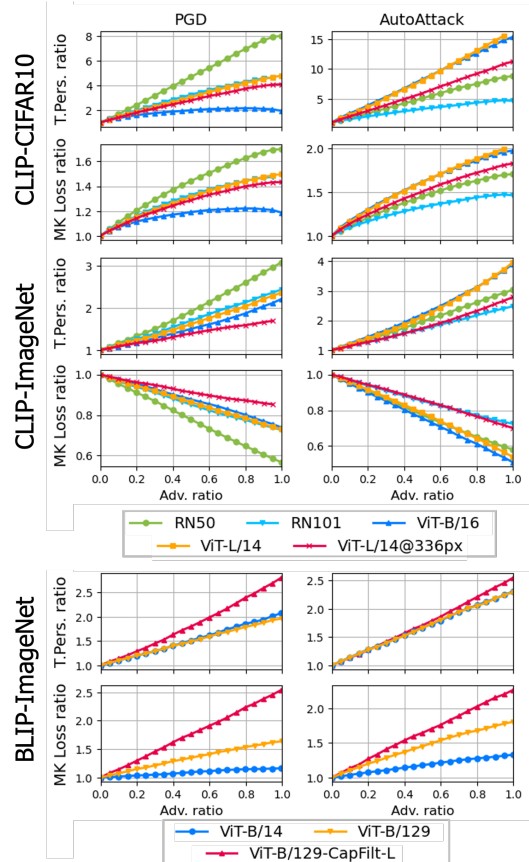

*Figure 3.* Normalized TP (row 1,3 and 5) and MK losses (row 2, 4 and 6) vs. the proportion of adversaries in the data batch ($\epsilon = 4/255$) in CLIP-CIFAR10, CLIP-ImageNet and BLIP-ImageNet.

that the topological signatures of the logits exhibit a consistent, monotonic change as the proportion of adversarial examples in the data batch increases.

**Experimental Setting:** Our analysis of multimodal adversaries is conducted on 10000 test samples extracted from CIFAR10, CIFAR100, and ImageNet datasets. The analysis is on 5 CLIP embedding models (ResNet50, ResNet101, ViT-B/16, ViT-L/14, and ViT-L/14@336px), 3 BLIP embedding models (ViT-B/14, ViT-B/129, and ViT-B/129-CapFilt-L) and 5 adversarial generation methods (FGSM, PGD, AutoAttack, APGD, and BIM). Due to space constraints, the complete description of the experiments, further results on CIFAR100, other CLIP embeddings, more attack methods, and attacking magnitudes are reported in Appx A.

**Behaviors of Adversarial Signatures:** For all test image samples, we fix the text and the BLIP/CLIP's alignment modules and generate adversarial images that successfully alter the zero-shot predictions. We then select only set of clean samples $S_X = \{x_i\}_{i=1}^N$ and the corresponding adversarial $S_{\tilde{X}} = \{\tilde{x}_i\}_{i=1}^N$, where the prediction on $x_i$ and $\tilde{x}_i$ are different. Starting from $S_X$, we iteratively replace $x_i$ by $\tilde{x}_i$ and obtain a mixture set $S_j^{mix} = \{x_i\}_{i=1}^j \cup \{\tilde{x}_i\}_{i=j+1}^N$

of clean and adversarial samples. Fig. 3 reports the impact of adversarial samples measured by the two proposed TP and MK losses as functions of the proportion of adversarial samples in a data batch, with an adversarial perturbation magnitude of $\epsilon = 4/255$. For better intuition, we report the normalized TP and MK losses, which are the TP/MK losses of $S_j^{mix}$, normalized by their respective values at $S_{j=0}^{mix} = S_X$. Thus, all plots start at 1.

*Table 1.* Monotonic behaviors of TP and MK for different models under adversarial attacks ($\epsilon = 4/255$). $(\uparrow, \downarrow)$ and $(\Uparrow, \Downarrow)$ indicate monotonically increasing/decreasing trends in TP and MK, respectively. The dash $-$ indicates non-monotonic behavior.

| Dataset | Model | PGD | AA | BIM | APGD | FGSM |
|---|---|---|---|---|---|---|
| CIFAR10 | CLIP RN50 | $\uparrow/\Uparrow$ | $\uparrow/\Uparrow$ | $\uparrow/\Uparrow$ | $\uparrow/\Uparrow$ | $\uparrow/\Uparrow$ |
|  | CLIP RN101 | $\uparrow/\Uparrow$ | $\uparrow/\Uparrow$ | $\uparrow/\Uparrow$ | $\uparrow/\Uparrow$ | $\uparrow/\Uparrow$ |
|  | CLIP ViT-B/32 | $-/-$ | $\uparrow/\Uparrow$ | $-/-$ | $\uparrow/\Uparrow$ | $\uparrow/-$ |
|  | CLIP ViT-L/14 | $\uparrow/\Uparrow$ | $\uparrow/\Uparrow$ | $\uparrow/\Uparrow$ | $\uparrow/\Uparrow$ | $\uparrow/-$ |
|  | CLIP ViT-L/14-336 | $\uparrow/\Uparrow$ | $\uparrow/\Uparrow$ | $\uparrow/\Uparrow$ | $\uparrow/\Uparrow$ | $\uparrow/-$ |
| ImageNet | CLIP RN50 | $\uparrow/\Downarrow$ | $\uparrow/\Downarrow$ | $\uparrow/\Downarrow$ | $\uparrow/\Downarrow$ | $-/\Downarrow$ |
|  | CLIP RN101 | $\uparrow/\Downarrow$ | $\uparrow/\Downarrow$ | $\uparrow/\Downarrow$ | $\uparrow/\Downarrow$ | $\uparrow/\Downarrow$ |
|  | CLIP ViT-B/32 | $\uparrow/\Downarrow$ | $\uparrow/\Downarrow$ | $\uparrow/\Downarrow$ | $\uparrow/\Downarrow$ | $-/\Downarrow$ |
|  | CLIP ViT-L/14 | $\uparrow/\Downarrow$ | $\uparrow/\Downarrow$ | $\uparrow/\Downarrow$ | $\uparrow/\Downarrow$ | $\uparrow/\Downarrow$ |
|  | CLIP ViT-L/14-336 | $\uparrow/\Downarrow$ | $\uparrow/\Downarrow$ | $\uparrow/\Downarrow$ | $\uparrow/\Downarrow$ | $\uparrow/\Downarrow$ |
| ImageNet | BLIP ViT-B/14 | $\uparrow/\Uparrow$ | $\uparrow/\Uparrow$ | $\uparrow/\Uparrow$ | $\uparrow/\Uparrow$ | $-/-$ |
|  | BLIP ViT-B/129 | $\uparrow/\Uparrow$ | $\uparrow/\Uparrow$ | $\uparrow/\Uparrow$ | $\uparrow/\Uparrow$ | $-/-$ |
|  | BLIP ViT-B/129-CL | $\uparrow/\Uparrow$ | $\uparrow/\Uparrow$ | $\uparrow/\Uparrow$ | $\uparrow/\Uparrow$ | $\downarrow/\Downarrow$ |

Fig. 3 illustrates the relationship between TP and MK losses as the proportion of adversarial samples in the data batch ($\epsilon = 4/255$) increases for CLIP-CIFAR10, CLIP-ImageNet, and BLIP-ImageNet. The important finding is that both losses generally exhibit monotonically changing behavior when clean samples are replaced with adversarial ones. Specifically, the TP loss steadily increases across nearly all experiments, while the MK loss increases with a higher proportion of adversarial samples in CLIP-CIFAR10 and BLIP-ImageNet but decreases consistently in CLIP-ImageNet. Despite these variations, both losses indicate that adversarial samples significantly alter the topological structure of the logits. In other words, although attacks make image samples align to different predicted texts, they do not preserve the topological structure of the adversarial images. Table 1 summarizes the monotonic behaviors of TP and MK losses across a wider range of attacks, offering an overview of the prevalence of these monotonic dynamics in more scenarios.

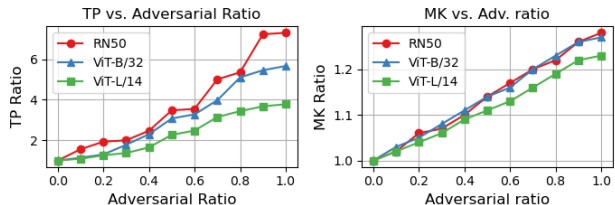

*Figure 4.* Normalized TP and MK losses vs. the proportion of text adversaries in the data batch in CLIP-ImageNet.

**Behaviors of Signatures of Text Adversarial:** Regarding

text-based adversarial attacks, we conducted experiments on *Cross-class prompt-injection* attacks in the text modality (Maus et al., 2023; Xu & Wang, 2024). Using three CLIP vision backbones (RN50, ViT-B/32, and ViT-L/14) on ImageNet, we gradually inject adversarial text prompts into each batch, track the resulting changes in TP and MK losses (Eqs. 3 and 5), and report the results in Fig. 4.

Similar to our image-based study, both the TP and MK losses change monotonically when the ratio of adversarial text increases, reinforcing the consistency of these topological measures across modalities.

### 3.3. Modeling Total Persistence of Adversaries

The final part of this section presents a theoretical explanation for the observed overall increase in TP of adversaries (see Table 1). Our assumption is that, since the primary goal of adversarial attacks is to change the top logit, they do not maintain the structure of the new label cluster. This leads to a more scattered and less organized representations. We call this the *adversarial scattering assumption*. Under that assumption, our hypothesis is that *the scattering behavior of adversarial logit leads to a higher TP*.

**PCP modeling:** To test the above hypothesis, we start with modeling the logits as points generated by a latent *Poisson Cluster Process (PCP)* (Daley & Vere-Jones, 2003), with clusters centered at the vertices of a $K$-dimensional simplex. For the following discussion, it is sufficient to understand that the PCP is governed by two parameters: $\alpha_s$ and a bias ratio $r$. Intuitively, a larger $r$ encourages the centers of clusters to be closer to the vertices of the simplex, and a larger $\alpha_s$ results in generated points being more concentrated around these centers. This relationship is visualized in Fig. 5. We provide a rigorous formulation of the PCP in Appx. B.

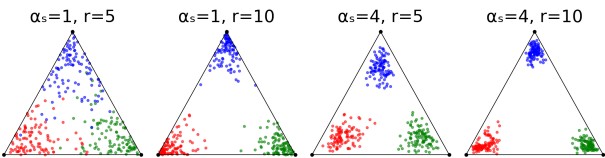

*Figure 5.* Points generated from PCP($\alpha_s, r$) in 2D-simplex.

**TP of PCP:** We utilize the PCP to investigate how the scattering of logits affects the TP. Due to the high complexity, we focus on *connected components* (0-dimensional homology), which are the most straightforward to analyze $\text{Pers}_0^\alpha(X) = \sum_{(b,d)\in D_0(X)} d^\alpha$, which is actually the length of the minimum spanning tree (MST) for the graph on $X$, and the edges corresponding to the birth-death pairs in $D_0(X)$ constitutes a MST in $X$ (Koyama et al., 2023). Thus, to determine $\text{Pers}_0^\alpha(X)$, we can alternatively focus on the MST and avoid the costly constructions of VR filtrations.

Although $\text{Pers}_0^\alpha(X)$ is the length of the MST, calculating it theoretically is still very challenging (Aldous & Steele, 1992), and the impact of the *scattering* of the data on the MST remain theoretically unclear. Consequently, we opt to use Monte Carlo simulations to investigate how the scattering of the logit impact its 0-th order TP $\text{Pers}_0^\alpha(X)$.

Fig. 6 illustrates the simulated MST length for 500 and 1,000 points in a 10-dimensional simplex as the parameters $\alpha_s$ and the ratio $r$ vary. In fact, when the PCP is more scattered (i.e., lower $\alpha_s$ and $r$), the length of the MST increases. *Thus, the adversarial scattering assumption implies the adversaries reduce the logits' concentration and results in a higher $\text{Pers}_0^\alpha(X)$. This is consistent with results in Sect. 3.2 and facilitates the use of TP to detect adversaries.*

**Higher-dimensional topological features:** During our research, our experimental results show that higher-degree summaries are less effective for the adversarial detection examined in this work. Specifically, we observe that critical distinctions between adversarial and non-adversarial logits primarily arise from lower-degree homology, particularly degree-0, rather than higher-degree topological features such as loops (degree-1) or cavities (degree-2).

This limitation of degree-1 homology can be explained via our PCP modeling. For example, as shown by the 3 configurations on the right of Fig. 5, when the filtration radius approaches half the distance between vertices of a simplex, a prominent one-dimensional hole emerges across all cases. As this degree-1 feature appears uniformly, it provides limited discriminative power among those configurations.

However, there are cases higher-order topological information can be beneficial. For example, Table 1 shows that FGSM exhibits different behaviors compared to more advanced attacks. We hypothesize that, because FGSM is relatively simple, it produces adversarial logits that is further from the target-class logits. Thus, instead of perturbing logits toward existing clean-data clusters—as more complex attacks do—FGSM creates entirely new clusters. Consequently, the mixture of clean and FGSM logits can have

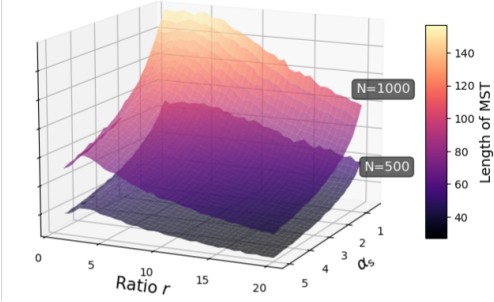

*Figure 6.* Monte Carlo simulations of the length of the MST ($\text{Pers}_0^\alpha(X)$) for different parameters of the PCP model.

twice as many clusters: one set from the clean data and one from the adversarial. As the adversarial ratio grows, the clean-data cloud first transforms into this *double-cluster* configuration—yielding a high TP—and then, at higher ratios, collapses into the adversarial cloud. This explains the peak in TP observed at intermediate mixture levels (see Fig. 11, 12 and 13, Appx. A). Although FGSM's persistence curve is not strictly monotonic, its distinctive, cluster-doubling behavior can be revealed via higher-order homology.

## 4. Maximum Mean Discrepancy with Topological Features

In this section, we demonstrate how to utilize the topological signatures identified in Sect. 3 for adversarial detection. Following (Gao et al., 2021; Grosse et al., 2017), we aim to address the *Statistical adversarial detection (SAD)* problem:

*Let $\mathcal{X} \subseteq \mathbb{R}^d$ and let $\mathbb{P}$ be a Borel probability measure on $\mathcal{X}$. Consider a dataset $S_X = \{x_i\}_{i=1}^n \sim \mathbb{P}^n$ composed of i.i.d. samples drawn from $\mathbb{P}$, and let $f : \mathbb{R}^d \to C$ denote the true labeling mapping for samples in $\mathbb{P}$, where $C$ is the set of labels. Suppose that adversaries have access to a classifier $\hat{f}$ trained on $S_X$ and i.i.d. samples $S'_X$ from $\mathbb{P}$. The objective is to determine whether a dataset $S_Y = \{y_i\}_{i=1}^m$ is originated from the distribution $\mathbb{P}$. We assume that $S_X$ and $S'_X$ are independent and no prior information about the attackers is available. $S_Y$ may consist of either i.i.d. samples from $\mathbb{P}$ or non-i.i.d. samples generated by attackers.*

In SAD, given a threshold $\alpha$, when $S_Y$ is drawn from $\mathbb{P}$, we want to accept the null hypothesis $H_0$ ($S_X$ and $S_Y$ are from the same distribution) with probability $1 - \alpha$. Conversely, if $S_Y$ includes adversarial samples, our objective is to reject $H_0$ with a probability approaching 1. It is important to note that the classifier $\hat{f}$ examined in our study is a zero-shot classifier based on multimodal alignment, rather than conventional feed-forward neural networks. We focus on the zero-shot problem because it is closely related to the alignment between modalities in multimodal systems.

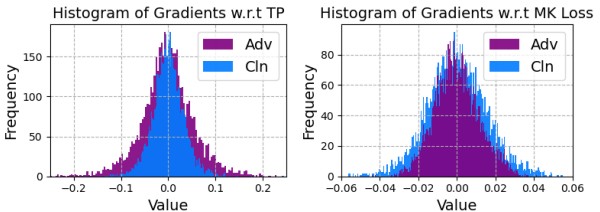

*Figure 7.* Topological Features of adversarial and clean inputs.

**Topological Features:** Our approach utilizing $\mathcal{L}_{TC}$ for SAD is to compute sample-level features derived from the topological loss $\mathcal{L}_{TC}$, i.e., the topological features are the gradients of the TC loss to each input features

$\dot{Y} = \nabla_Y \mathcal{L}_{TC}(Y, T)$, where $Y$ represents the image's logits and $T$ denotes the text embedding. Intuitively, the gradients capture how each image's feature contributes to the changes in the topological image-text alignment. The gradients are computed by back-propagating Eq. 3 and 5 via Pytorch's implementations of the homologies (AidosLab, 2023).

To ensure the independence assumption in SAD, the topological features need to be computed independently. This can be achieved by utilizing a hold-out dataset $Z$, i.e., $\dot{y}_i = \nabla_{y_i} \mathcal{L}_{TC}(\{y_i\} \cup Z, T), \forall y_i \in S_Y$. However, this requires a VR filtration for each $y_i \in S_Y$, which is computationally prohibitive. Instead, our implementation adopts a batch-processing approach:

$$\dot{Y} = \nabla_Y \mathcal{L}_{TC}(Y \cup Z, T) \tag{6}$$

While this approach does not ensure independence, it only need to construct a single VR filtration. The introduced error is acceptable when $|Z|$ is significantly larger than $|Y|$. The above process is illustrated by the backward arrows of Fig. 2 and detailed in Appx. C (Alg. 1).

As an illustration, Fig. 7 displays histograms of the exact gradients, computed for 500 different $y_i$ in the CIFAR10 dataset, where $|Z| = 1000$. It shows that distinct signatures of adversaries are reflected in our extracted topological features, suggesting high potential for more accurate detection.

**Topological MMD:** We now describe how to utilize the extracted topological features to improve the SOTA detection method SAMMD (Gao et al., 2021). SAMMD employs the *semantic-aware deep kernel* $k_\omega(x_{\text{in}}, y_{\text{in}})$ to evaluate the similarity between input images (Detailed in Appx. D). To incorporate topological features, we modify SAMMD's kernel $k_\omega$ used for Eq. 1 and propose the following topological-contrastive deep kernel $k_\tau$:

$$k_\tau(x_{\log}, y_{\log}) = \left[(1 - \epsilon_0) \tau_{\hat{f}}(x_{\log}, y_{\log}) + \epsilon_0\right] \nu_{\hat{f}}(x_{\log}, y_{\log})$$

Unlike $k_\omega$, our kernel operates on the logits $x_{\log}$ and $y_{\log}$, instead of the input. The term $\nu_{\hat{f}}(x_{\log}, y_{\log}) = \kappa(x_{\log}, y_{\log})$ evaluates the similarity between $x_{\log}$ and $y_{\log}$ using the image embeddings extracted from the layer before the alignment with the text embedding. On the other hand, $\tau_{\hat{f}}(x_{\log}, y_{\log}) = \kappa_{\text{TC}}(\dot{x}, \dot{y})$ measures the similarity between the topological features $\dot{x}$ and $\dot{y}$ obtained from Eq. 6. Here, $\kappa$ and $\kappa_{\text{TC}}$ are Gaussian kernels with bandwidth $\sigma$ and $\sigma_{\text{TC}}$. Using TP or MK loss produces different features $\dot{x}$ and $\dot{y}$, resulting in the methods Total Persistence SAMMD (TP-SAMMD) and Multi-scale Kernel SAMMD (MKSAMMD).

With $k_\tau$, the discrepancy between clean and adversarial data is estimated using the $U$-statistic estimator $\frac{1}{n(n-1)} \sum_{i,j=1, i\neq j}^n H_{ij}$ where $H_{ij} = k_\tau(x_{\log_i}, x_{\log_j}) + k_\tau(y_{\log_i}, y_{\log_j}) - 2 k_\tau(x_{\log_i}, y_{\log_j})$. Similar to SAMMD, the parameters of TPSAMMD and MKSAMMD are optimized via gradient descents.

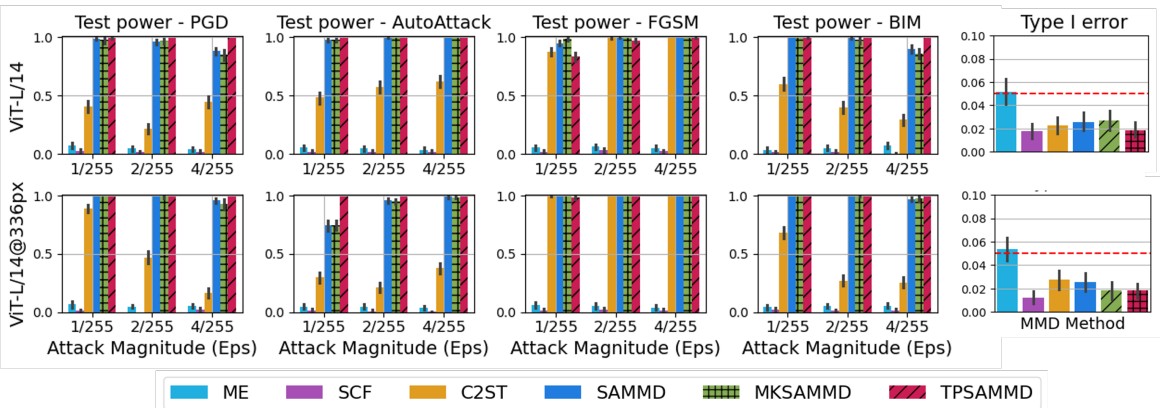

*Figure 8.* Test power and average Type-I error (last column) of adversarial detection methods in CIFAR10 with CLIP embedding.

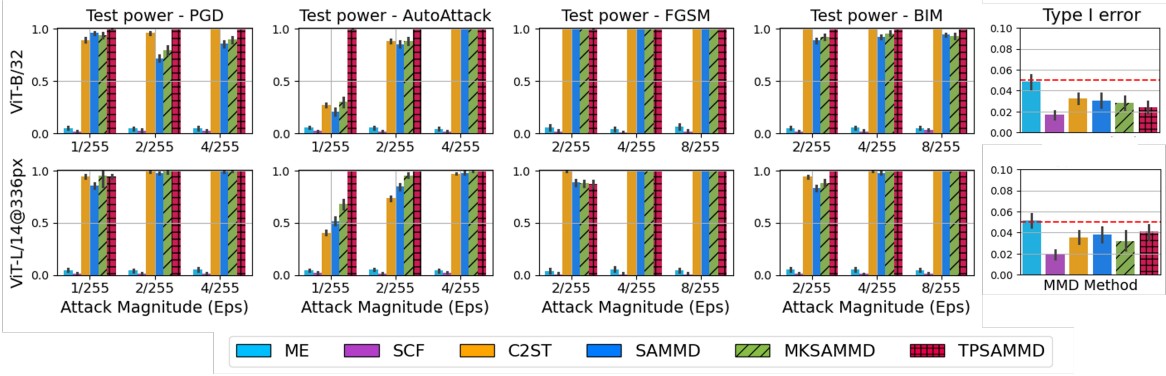

*Figure 9.* Test power and average Type-I error (last column) of adversarial detection methods in ImageNet with CLIP embedding.

## 5. MMD Experimental Results

**Settings and baselines:** We validate the advantage of topological features for MMD in the settings stated in Subsect. 3.2. We compare TPSAMMD and MKSAMMD tests with 4 existing two-sample tests: 1) SAMMD (Gao et al., 2021), 2) Mean Embedding (ME) test (Jitkrittum et al., 2016); 3) Smooth Characteristic Functions (SCF) test (Chwialkowski et al., 2015); and 4) Classifier two-sample test (C2ST) (Liu et al., 2020).

Each MMD test is conducted on two disjoint subsets of clean and adversarial samples, each containing 50 images for CIFAR10 and CIFAR100, and 100 images for ImageNet. The sizes of the holdout data $Z$ for the topological features computation (Eq. 6) are 1000 and 3000 for CIFAR10/100 and ImageNet, respectively. Those sizes of the holdout are selected to ensure at least some samples per class are represented in the point cloud. Notably, this chosen setting is significantly more challenging than existing experiments (Gao et al., 2021), which differentiate between sets of 500 samples. Our additional experiments in Fig. 17 Appx. E illustrates the above claim by reporting the impact of the number of MMD's samples and $|Z|$ on the performance of our proposed methods. Each test was conducted over

100 trials with Type-I error controlled at $\alpha = 0.05$. Due to the page limit, results on CIFAR100 and more in-depth experimental results are provided in Appx. E.

**CIFAR10:** Fig. 8 reports the test power of adversarial detection methods for CLIP alignment in CIFAR10. Overall, the SAMMD-based methods achieve almost $100\%$ test power across many attack types, even at very small noise magnitudes. Nevertheless, TPSAMMD shows a slight advantage in PGD and AA. Additionally, we average the Type-I errors for all tests in the first 4 columns at $\epsilon = 4/255$ and report them in the last column. It shows that all methods keep the error controlled at $5\%$.

**ImageNet:** Fig. 9 and 10 report the detection test power for ImageNet with CLIP and BLIP, respectively. The results demonstrate that, while MKSAMMD stays comparative to SOTA, our proposed TPSAMMD outperforms existing baselines in terms of test power, showcasing the advantage of integrating topological features in distinguishing adversarial samples from clean ones. Notably, the difference in test power between our approach and traditional methods becomes more pronounced as the adversarial perturbation is small, highlighting the robustness of our detection mechanism under more challenging attack scenarios. Similar to

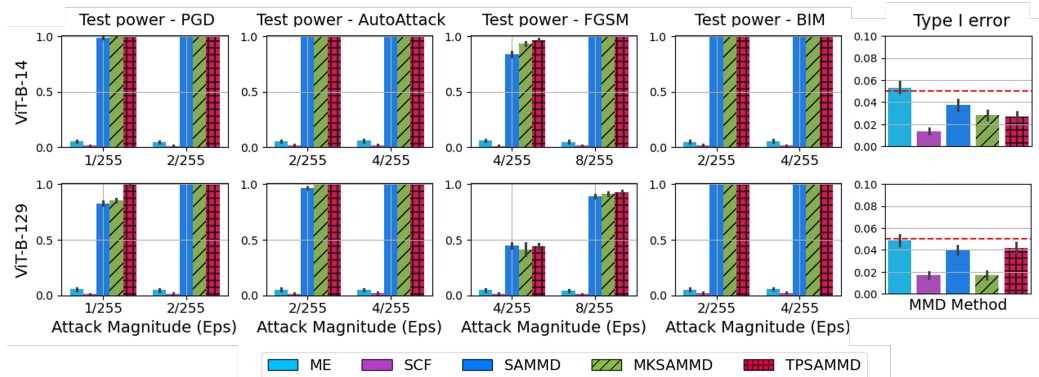

*Figure 10.* Test power and average Type-I error (last column) of adversarial detection methods in ImageNet with BLIP embedding.

CIFAR10, the Type-I errors of all methods in ImageNet, except ME, remain strictly below $5\%$.

*Table 2.* Gain in accuracy of TPSAMMD vs SAMMD in PGD.

| $\epsilon$ | CLIP ViT B/32 | CLIP ViT-L/14 | BLIP ViT-B/14 | BLIP ViT-B/129 |
|---|---|---|---|---|
| 1/255 | 3.6% | 8.6% | 1.5% | 16.7% |
| 4/255 | 14.3% | 8.2% | 0% | 0% |

**Gain of Topological Feature:** Table 2 highlights the accuracy gains of TPSAMMD compared to SAMMD under PGD attacks. For BLIP at $\epsilon = 4/255$, the gain is 0% because both methods achieve 100% test power. Additionally, the inclusion of topological features clearly enhances the MMD test. This demonstrates that by incorporating information on how adversarial logits disrupt the overall topological structure of the data into the detection process, we can substantially improve detection capabilities.

## 6. Conclusion and Future Work

This study studies the vulnerability of multimodal ML systems, such as CLIP and BLIP, to adversarial attacks by exploring the topological disruptions in text-image alignment. We introduced two novel TC losses to identify distinctive signatures of adversaries, and demonstrate that these losses exhibit consistent monotonic changes across various attacks. By integrating them into MMD tests, we developed new adversarial detection methods that significantly enhance detection accuracy. This approach not only deepens our understanding of multimodal adversarial attacks but also provides practical tools to strengthen their resilience.

Future work will explore these topological analysis to other multimodal configurations, and demonstrate the potential of topological methods in enhancing the robustness and reliability of multimodal ML systems.

## Acknowledgement

This manuscript has been assigned LA-UR-25-20561. This research was funded by the Los Alamos National Laboratory (LANL) Laboratory Directed Research and Develop- ment (LDRD) program under grants 20230287ER & 20240868PRD3 and supported by LANL's Institutional Computing Program, and by the U.S. Department of Energy National Nuclear Security Administration under Contract No. 89233218CNA000001.

## Impact Statement

This work addresses a gap in the defense of multimodal ML systems against adversarial attacks. By introducing novel Topological-Contrastive losses and leveraging persistent homology, this study not only deepens our understanding of how adversarial attacks disrupt text-image alignment but also provides practical methods to detect such disruptions with high accuracy. As Multimodal ML systems are increasingly used in sensitive domains, enabling more precise detection will enhance the security and resilience of systems that directly impact public safety, health, and trust in technology. Furthermore, the monotonic properties of the proposed losses provide interpretable insights into adversarial behavior, promoting transparency and trustworthiness in adversarial defense strategies.

While this work provides powerful tools to detect and mitigate adversarial threats, it also necessitates careful ethical considerations. The methods developed here could potentially be misused to generate more advanced adversarial attacks, exacerbating the arms race between attackers and defenders. To mitigate this risk, we advocate for responsible dissemination of these techniques and call for collaboration across the research community to ensure their ethical application. Future directions include extending these topological methods to other multimodal configurations, such as video-text and audio-text systems, further solidifying the role of topological insights in enhancing the robustness of multimodal ML systems. This research demonstrates the transformative potential of topology-based approaches in securing the next generation of ML systems and sets a strong foundation for future advancements in the field.

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

## A. Experimental settings and results on Total Persistence and Multi-scale Kernel

Our experiments were conducted on a cluster with nodes featuring four NVIDIA Hopper (H100) GPUs each, paired with NVIDIA Grace CPUs via NVLink-C2C for rapid data transfer essential for intensive computational tasks. Each GPU is equipped with 96GB of HBM2 memory, ideal for handling large models and datasets.

We evaluated five CLIP models, six attack methods, and three datasets. We employed torch-attack (Kim, 2020) to generate adversarial perturbations with magnitudes $\epsilon$ of $\frac{1}{255}$, $\frac{2}{255}$, $\frac{4}{255}$, and $\frac{8}{255}$. The Square attack for ImageNet was excluded due to its high computational complexity, which hinders generating a sufficiently large dataset for reliable topological data analysis. Comprehensive results are presented in Figs 11, 12, 13, 14, and 15. Specifically, we report the TP and MK losses as defined in Eq.3 and Eq.5, respectively, against the adversary ratio. For each experiment, we processed all 10,000 test samples to generate adversarial examples, retaining only those that successfully disrupted CLIP alignment. Starting with the full set of clean samples, we randomly replaced samples with adversaries according to each adversarial ratio (x-axis), computed the corresponding TC losses, and presented the results in the figures.

Unlike the main manuscript, which presents results only for an attack magnitude of $\epsilon = 4/255$, we provide comprehensive results across various models and attack magnitudes. Except for FGSM, the TC loss shows monotonic behavior across different models, attack methods, and magnitudes. We hypothesize that FGSM's simplicity leads to fewer successful samples and larger distortions, resulting in unrealistic samples and disrupting the original point clouds differently than more sophisticated attacks. This simplicity affects the statistical outcomes of our computations, distinguishing FGSM from more refined attack methods. Additionally, we observe a trend that warrants future research: more sophisticated attack methods and advanced models exhibit more pronounced monotonic behavior in TC losses. This is evident when comparing TC losses of the ViT-L family to ViT-B, ResNet101 to ResNet50, and PGD/AA to FGSM.

## B. Modeling Logits as Poisson Cluster Process

In Subsect. 3.3, we propose to use Poisson Cluster Process to model the logits resulting from the embedding modules in multimodal alignments. We now provide the details formulation of our PCP modeling.

**Poisson Cluster Process:** Our PCP modeling is constructed from a *Parent Simplex*. Then, the actual point cloud is generated by sampling the *Children Clusters* surrounding the vertices of that simplex. In particular, the process is described as follows:

- *Parent Simplex*: A $K$-dimensional simplex is the convex hull of $K + 1$ affinely independent points (called *parents points*) in $\mathbb{R}^K$. Let those parents points of the simplex be $V = \{v_0, v_1, \ldots, v_K\}$, the simplex is given as:

$$S = \left\{ x \in \mathbb{R}^K \;\middle|\; x = \sum_{i=0}^{K} \lambda_i v_i, \; \lambda_i \geq 0, \; \sum_{i=0}^{K} \lambda_i = 1 \right\}$$

  Any point $x$ inside the simplex is a convex combination of its vertices.

- *Children Clusters*: The logits are modeled as points clustering around the simplex vertices. For vertex $v_i$, $N_i$ points are generated using coefficients $\lambda_i$ sampled from a Dirichlet distribution parameterized by $\boldsymbol{\alpha}_i = \{\alpha_{i,j}\}_{j=0}^{K}$:

$$f_{\boldsymbol{\alpha}_i}(\lambda_0, \lambda_1, \ldots, \lambda_K) = \frac{1}{\mathrm{B}(\boldsymbol{\alpha}_i)} \prod_{j=0}^{K} \lambda_k^{\alpha_{i,j}-1}$$

  where $\mathrm{B}(\boldsymbol{\beta})$ is the multivariate Beta function $\mathrm{B}(\boldsymbol{\beta}) = \prod_{j=0}^{K} \Gamma(\beta_j)/\Gamma\left(\sum_{j=0}^{K} \beta_j\right)$ and $\Gamma(\cdot)$ is the gamma function. Each generated point $x$ is then computed as $x = \sum_{i=0}^{K} \lambda_i v_i$.

This construction captures the clustering behavior of logits of $K + 1$ labels and provides a basis for understanding how adversarial perturbations might increase topological complexity by introducing distortions in the cluster distribution. Intuitively, for each vertex $i$, a larger $\alpha_{i,i}$ and smaller $\alpha_{i,j}(j \neq i)$ increases the probability that $\lambda_i$ is close to 1 and encourages points of cluster $i$ stay nearer to $v_i$. Additionally, a larger overall $\alpha_{i,j}$ will increase the concentration of $\lambda$ at the

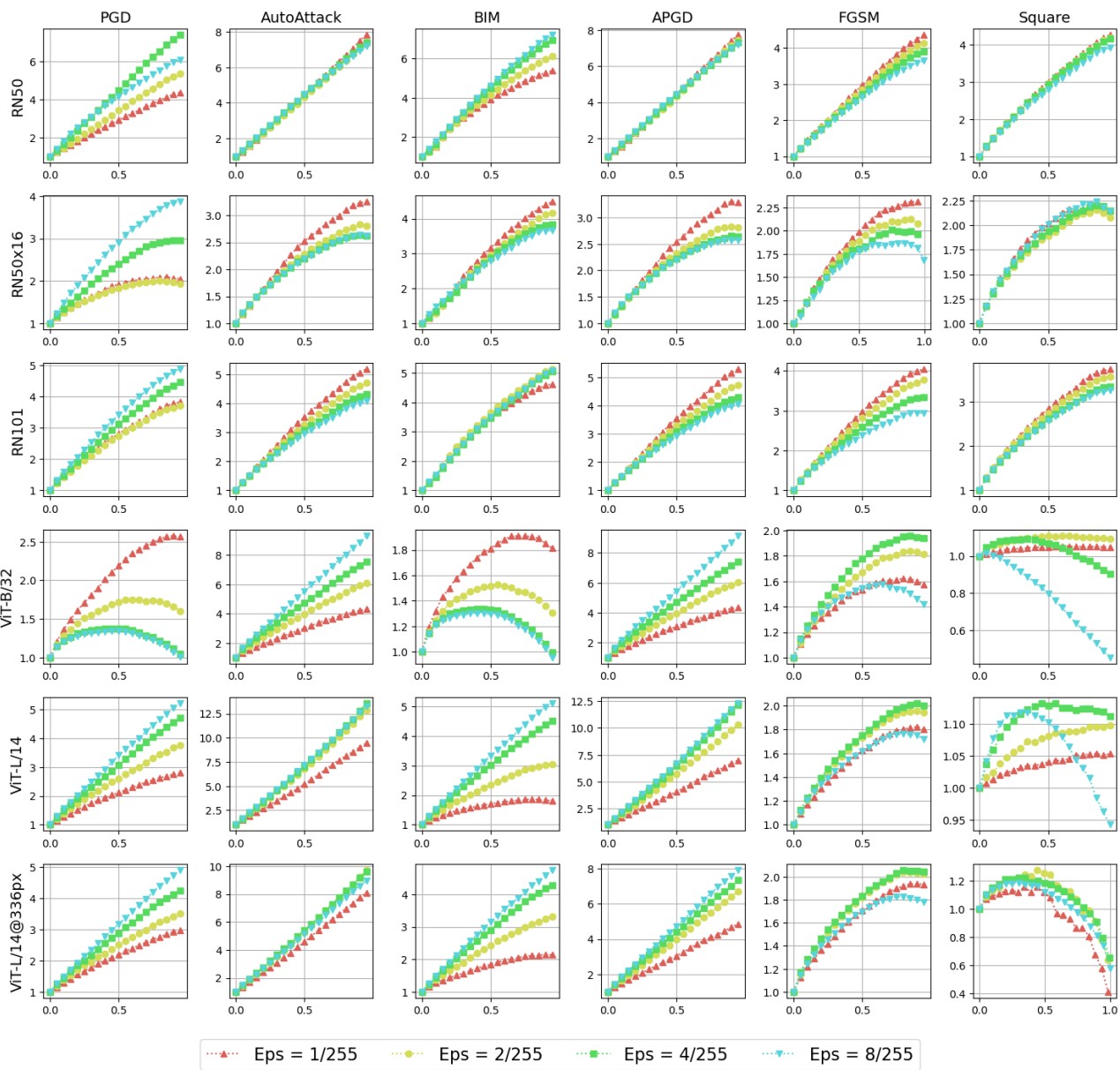

*Figure 11.* Normalized Total Persistence versus Adversarial data proportions in CIFAR10.

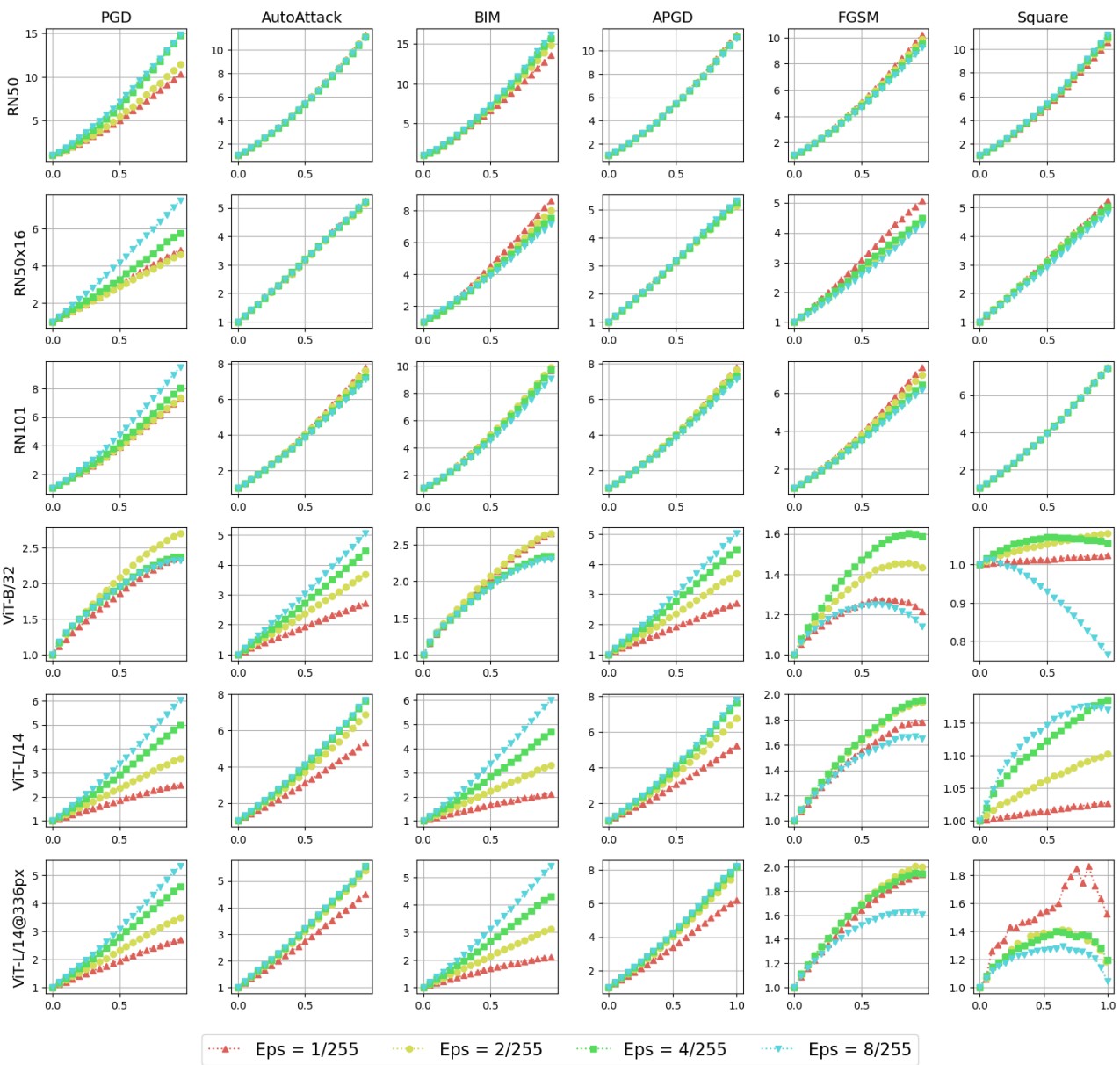

*Figure 12.* Normalized Total Persistence versus Adversarial data proportions in CIFAR100.

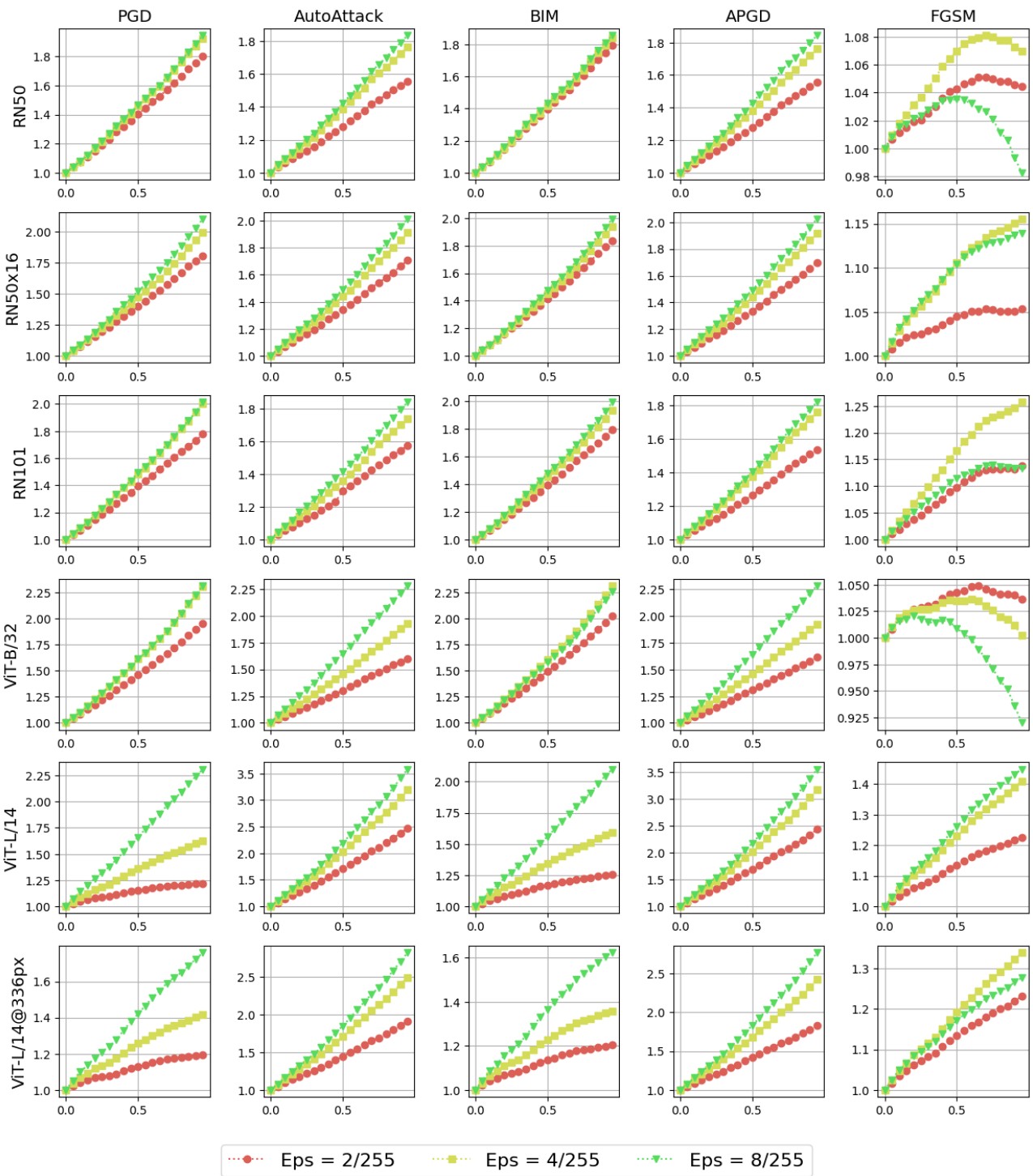

*Figure 13.* Normalized Total Persistence versus Adversarial data proportions in ImageNet.

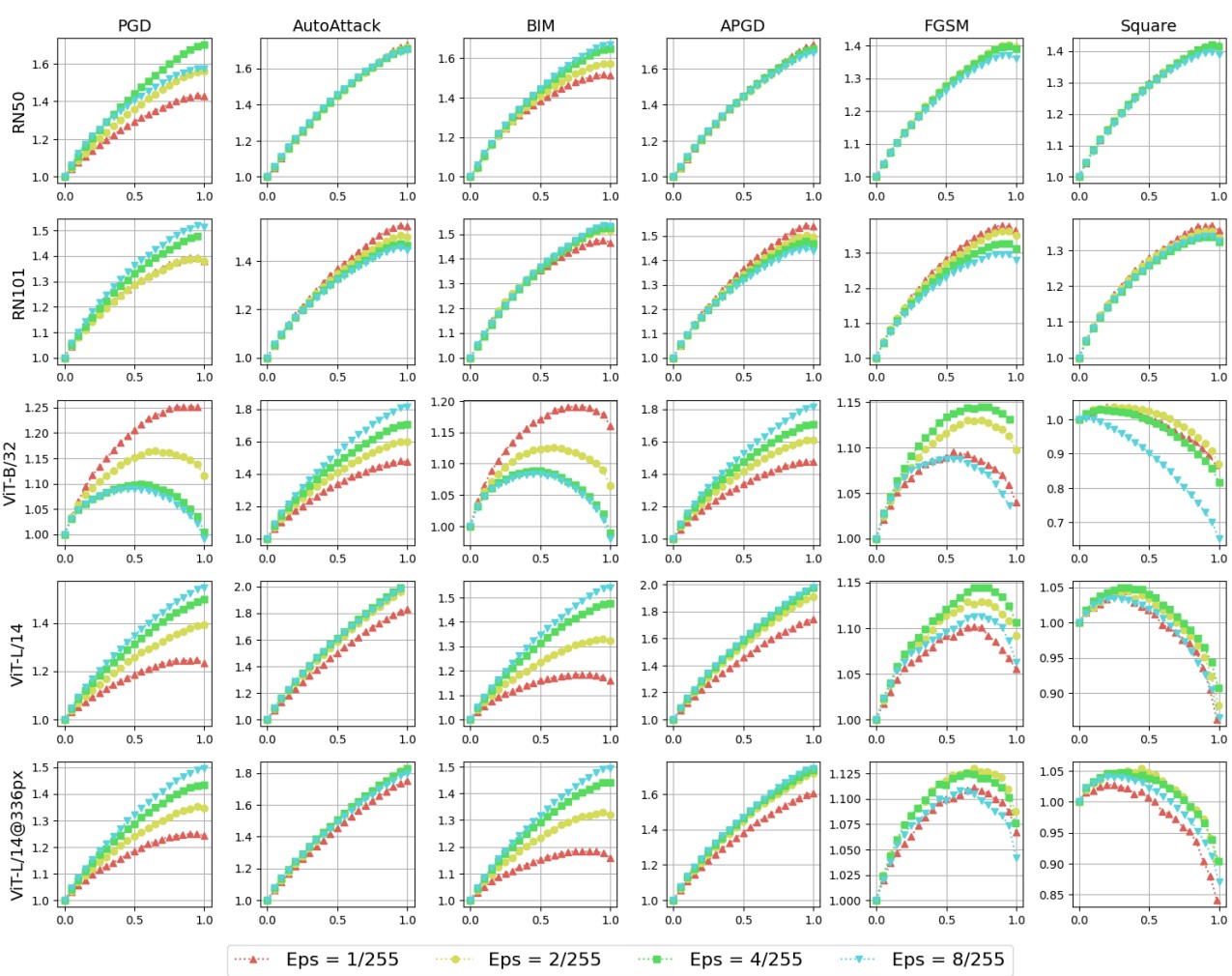

*Figure 14.* Normalized Multi-scale Kernel loss versus Adversarial data proportions in CIFAR10.

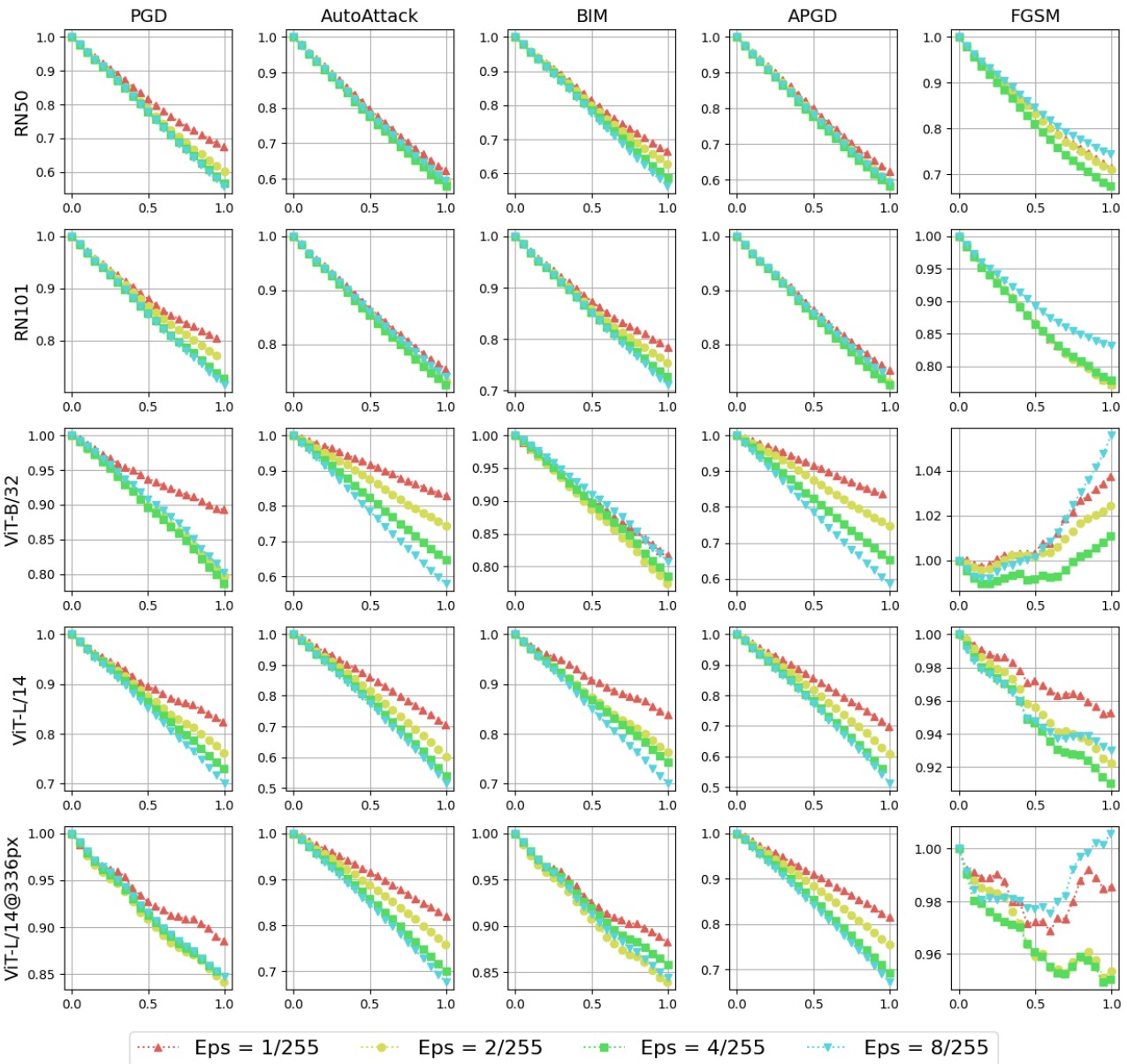

*Figure 15.* Normalized Multi-scale Kernel loss versus Adversarial data proportions in ImageNet.

cluster's center. For the sake of brevity, we set $\alpha_{i,i} = \alpha_{\text{large}}$ and $\alpha_{i,j(j \neq i)} = \alpha_{\text{small}}$. This gives us the mean and variance of $\lambda$:

$$\mathbb{E}[\lambda_i] = \frac{\alpha_{\text{large}}}{\alpha_{\text{total}}}, \quad \mathbb{E}[\lambda_{j(j \neq i)}] = \frac{\alpha_{\text{small}}}{\alpha_{\text{total}}}$$

$$\text{Var}[\lambda_i] = \frac{\tilde{\alpha}_{\text{large}}(1 - \tilde{\alpha}_{\text{large}})}{\alpha_{\text{total}} + 1}, \quad \text{Var}[\lambda_{j(j \neq i)}] = \frac{\tilde{\alpha}_{\text{small}}(\alpha_{\text{total}} - \tilde{\alpha}_{\text{small}})}{\alpha_{\text{total}} + 1} \tag{7}$$

where $\alpha_{\text{total}} = \sum_{j=0}^{K} \alpha_{i,j}$, $\tilde{\alpha}_{\text{large}} = \alpha_{\text{large}}/\alpha_{\text{total}}$ and $\tilde{\alpha}_{\text{small}} = \alpha_{\text{small}}/\alpha_{\text{total}}$. The parameter $\alpha_s$ and $r$ in Fig. 5 and the main manuscript are the $\alpha_{\text{small}}$ and $1/\tilde{\alpha}_{\text{small}}$ mentioned in this appendix.

**Modeling adversaries as PCP:** Our adversarial scattering assumption (stated in the main manuscript) posits that adversarial attacks primarily target the top logit component, causing logits to shift near different clusters without preserving the new cluster's structure. This leads to more scattered and disorganized point clouds. Consequently, we hypothesize that when label clusters of clean and adversarial point clouds are modeled as PCPs, the PCP for clean data will exhibit greater concentration and lower variance compared to that of adversarial data.

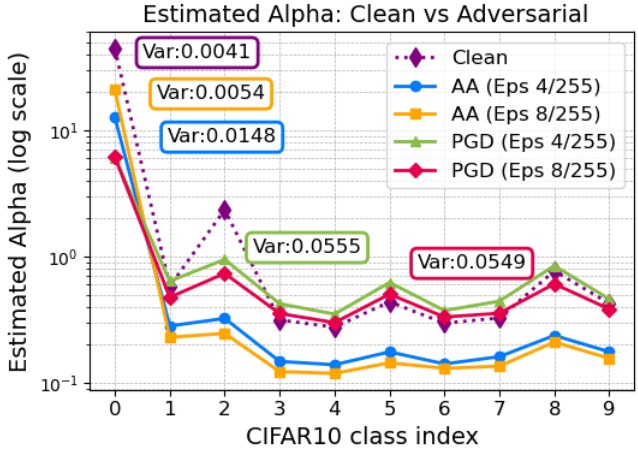

*Figure 16.* MLE of Dirichlet's $\alpha$ coefficients for different logits: the MLE of clean logits results in the Dirichlet distribution with the lowest variance.

We empirically validate the aforementioned assumption through the experiments presented in Fig. 16. Specifically, we collect both clean and adversarial logits (generated using AA and PGD) for class index 0 from the CLIP-ViT-L/14@336px model applied to the CIFAR-10 dataset with attacking levels $\epsilon \in \{4/255, 8/255\}$. We then employ Maximum Likelihood Estimation to fit the PCP's $\alpha$ parameter. As expected, the $\alpha_0$ component attains the highest value. Notably, the fitted PCP for clean samples exhibits the highest concentration, characterized by the largest $\alpha_0$ and overall $\alpha_{i=1,\ldots,9}$. Additionally, we report the corresponding variances of the PCP models fitted to these data, which indicate that adversarial data result in logits with higher variance.

## C. Computation of Topological Feature

Algorithm 1 outlines the pseudocode for computing the TP and MK losses. Given a point cloud $Y$ and a reference point cloud $T$, the algorithm calculates the topological feature $\dot{Y} = \nabla_Y \mathcal{L}_{TC}(Y \cup Z, T)$ as defined in Eq. 6. In our MMD test, $Y$ and $Z$ represent the logits or embeddings of the examined images (both clean and adversarial) and a hold-out image from the clean data, respectively. On the other hand, $T$ denotes the embeddings of the label classes for the corresponding datasets: the 1000 class labels of ImageNet, 100 class labels of CIFAR100, or 10 class labels of CIFAR10.

## D. Semantic-Aware Maximum Mean Discrepancy

Although $\text{MMD}(\mathbb{P}, \mathbb{Q}; \mathcal{H}_k)$ is a perfect statistic to check if $\mathbb{P}$ and $\mathbb{Q}$ are the same, its empirical test power depends significantly on the used kernels (Sutherland et al., 2016; Liu et al., 2020). Semantic-Aware Maximum Mean Discrepancy is a MMD test introduced by (Gao et al., 2021), which utilizes the following kernel acting on *semantic features* extracted by a

---

**Algorithm 1** Topological-Contrastive Feature Extraction

---

**Input**: Examined point cloud $Y$, and reference point cloud $T$
**Params**: *method* (*Total persistence* or *Multi-kernel*), hold-out data $Z$, order $\alpha$, and homology dimension $K$
**Output**: Topological-Contrastive Features $\nabla_Y \mathcal{L}_{TC}(Y \cup Z, T)$

1:  Construct the Vietoris–Rips complexes $\{\mathfrak{R}_\epsilon(Y \cup Z)\}_\epsilon$ and $\{\mathfrak{R}_\epsilon(T)\}_\epsilon$
2:  **for** $i$ from 0 to $K$ **do**
3:      Construct $D_i(Y \cup Z)$ from $\{\mathfrak{R}_\epsilon(Y \cup Z)\}_\epsilon$
4:      Construct $D_i(T)$ from $\{\mathfrak{R}_\epsilon(T)\}_\epsilon$
5:      **if** *method* $==$ *Total persistence* **then**
6:          $\mathrm{Pers}_i^\alpha(Y \cup Z) := \sum_{(b,d) \in D_i(Y \cup Z)} (d - b)^\alpha$
7:          $\mathrm{Pers}_i^\alpha(T) := \sum_{(b,d) \in D_i(T)} (d - b)^\alpha$
8:      **else if** *method* $==$ *Multi-kernel* **then**
9:          $k_i := k_\sigma(D_i(Y \cup Z), D_i(T))$
10:     **end if**
11: **end for**
12: **if** *method* $==$ *Total persistence* **then**
13:     $\mathcal{L}_{TC}(Y \cup Z, T) = \| \sum_{i=0}^K \mathrm{Pers}_i^\alpha(Y \cup Z) - \mathrm{Pers}_i^\alpha(T) \|_\alpha$
14: **else if** *method* $==$ *Multi-kernel* **then**
15:     $\mathcal{L}_{TC}(Y \cup Z, T) = \sum_i^K k_i$
16: **end if**
17: Back-propagate TC Loss: $\dot{Y} = \nabla_X \mathcal{L}_{TC}(Y \cup Z, T)$
18: **return** $\dot{Y}$

---

well-trained classifier on clean data:

$$k_\omega(x_{\text{input}}, y_{\text{input}}) = \left[ (1 - \epsilon_0)\, s_{\hat{f}}(x_{\text{input}}, y_{\text{input}}) + \epsilon_0 \right] q(x_{\text{input}}, y_{\text{input}}),$$

where $s_{\hat{f}}(x_{\text{input}}, y_{\text{input}}) = \kappa\big(\phi_{\hat{f}}(x_{\text{input}}), \phi_{\hat{f}}(y_{\text{input}})\big)$ is a deep kernel function that assesses the similarity between $x_{\text{input}}$ and $y_{\text{input}}$ using features extracted from the last fully connected layer of the classifier $\hat{f}$. Here, $\kappa$ denotes the Gaussian kernel with bandwidth $\sigma_{\phi_{\hat{f}}}$, $\epsilon_0 \in (0, 1)$, and $q(x_{\text{input}}, y_{\text{input}})$ represents the Gaussian kernel with bandwidth $\sigma_q$. Given that kernel, the SAMMD is proposed to measure the discrepancy between natural and adversarial data:

$$\mathrm{SAMMD}(\mathbb{P}, \mathbb{Q}) = \mathbb{E}\left[ k_\omega(X, X') + k_\omega(Y, Y') - 2k_\omega(X, Y) \right],$$

where $X, X' \sim \mathbb{P}$ and $Y, Y' \sim \mathbb{Q}$. The $U$-statistic estimator used to empirically approximate $\mathrm{SAMMD}(\mathbb{P}, \mathbb{Q})$ is given by:

$$\mathrm{SAMMD}^2(S_X, S_Y; k) = \frac{1}{n(n-1)} \sum_{i \neq j} H_{ij},$$

where

$$H_{ij} = k_\omega(x_i, x_j) + k_\omega(y_i, y_j) - k_\omega(x_i, y_j) - k_\omega(y_i, x_j).$$

## E. MMD Experimental Results

This appendix provides the complete results of our Maximum Mean Discrepancy (MMD) experiments on ImageNet (Fig. 18 and 19), CIFAR-10 (Fig. 20 and 21), and CIFAR-100 (Fig. 22 and 23). Specifically, we report the test power and Type-I error rates of various methods for detecting adversarial samples in batches. Each test batch consists of 50 clean or adversarial samples. Fig. 17 demonstrate the impact of the number of TST samples and the holdout size on the performance of our proposed TPSAMMD method in CLIP-RN50 in ImageNet dataset. We can clearly see that our method can perform well even with $|Z|$ slightly more than 1000 samples.

Notably, this setting is significantly more challenging than existing works (Gao et al., 2021), which use test batches of size 500, at which TPSAMMD constantly achieve test power of 1.0 (Fig. 17). As a result, several methods, such as ME and SCF, perform considerably worse than previously reported. We chose this more demanding setting because the state-of-the-art

method, SAMMD, already achieves 100% test power with 500-sample batches. By reducing the batch size to 50, we emphasize the advantages of our topological features in a more challenging scenario.

The results demonstrate that SAMMD-related methods achieve highly competitive test power across a wide range of models and attack methods. This outcome aligns with previous findings in unimodal settings (Gao et al., 2021). This consistency motivated us to integrate our topological features into SAMMD, showcasing the advantages of our approach in practical scenarios. As shown, TPSAMMD consistently performs as well as or better than SAMMD in nearly all settings, particularly against PGD and AA attacks and within more complex Vision Transformer models. On the other hand, although MKSAMMD performs slightly worse than TPSAMMD, it still offers improvements over SAMMD in many configurations.

The code used in this study is currently under review for release by the organization. We are awaiting approval, and once granted, the code will be made publicly available.

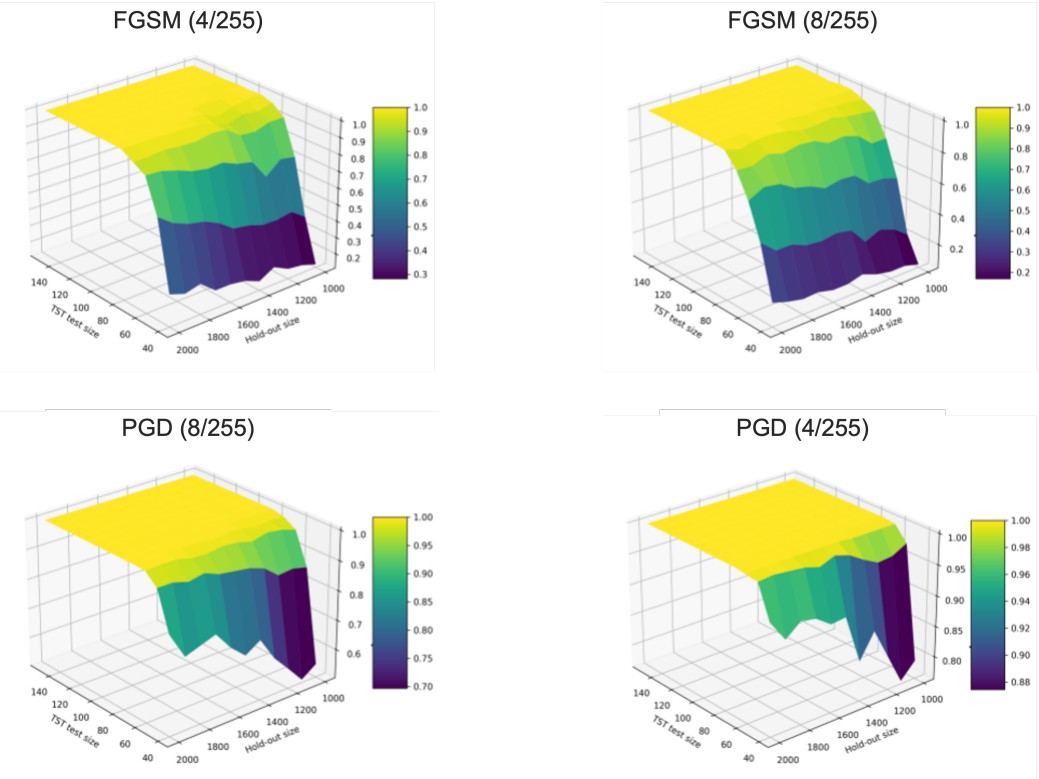

*Figure 17.* Test power of TPSAMMD in CLIP-RN50 with FGSM and PGD with different hold-out dataset sizes and different TST test size.

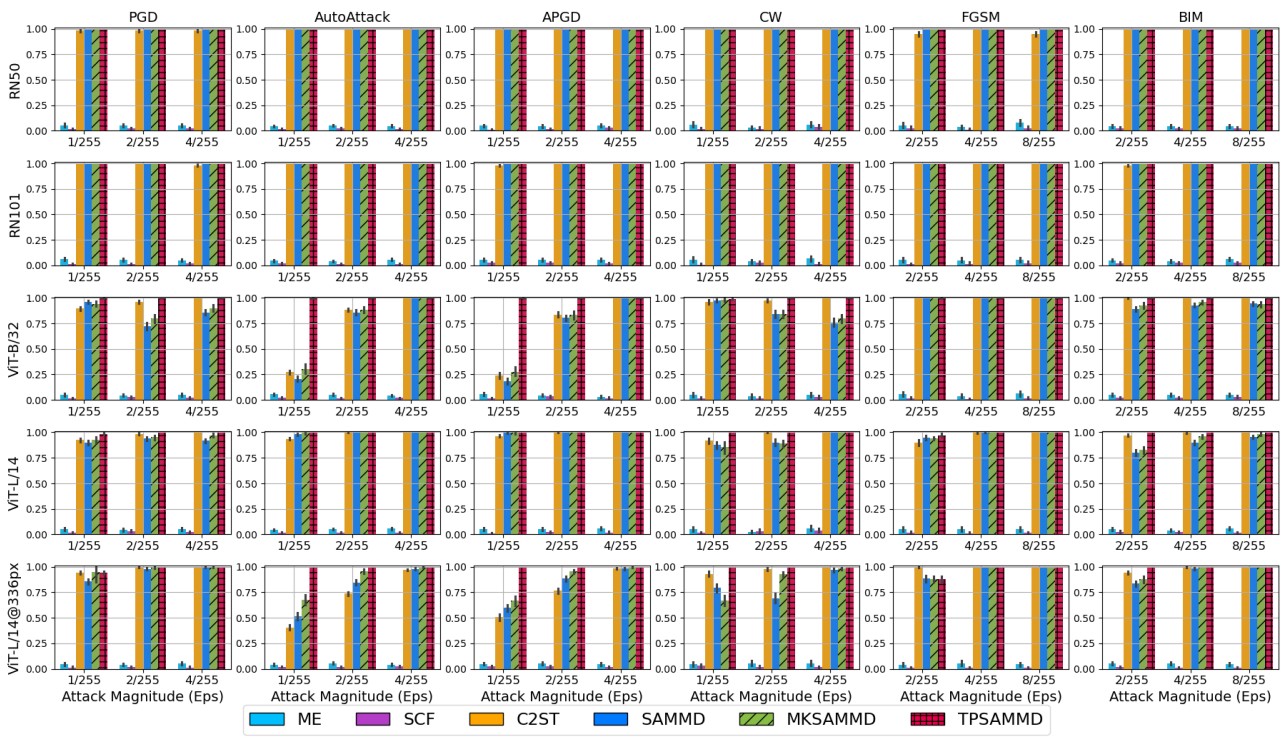

*Figure 18.* Test power of different methods in detecting ImageNet's adversaries.

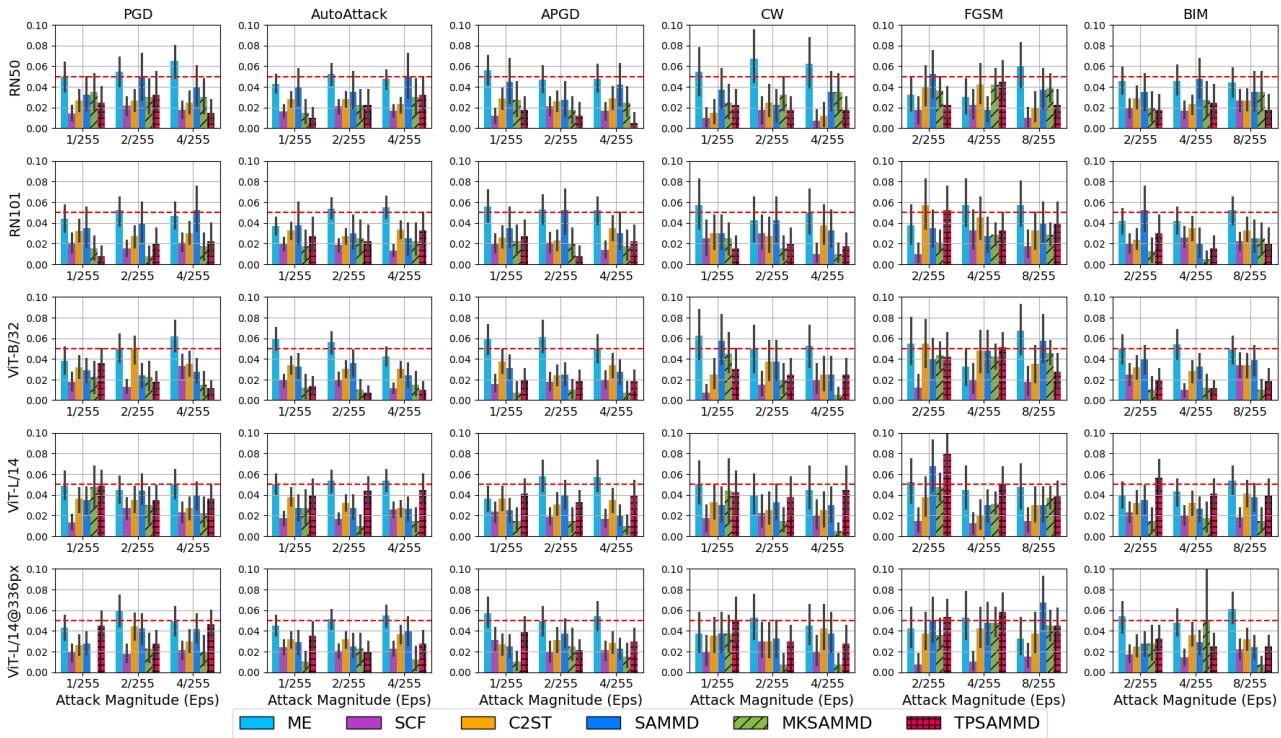

*Figure 19.* Type-I error of different methods in detecting ImageNet's adversaries.

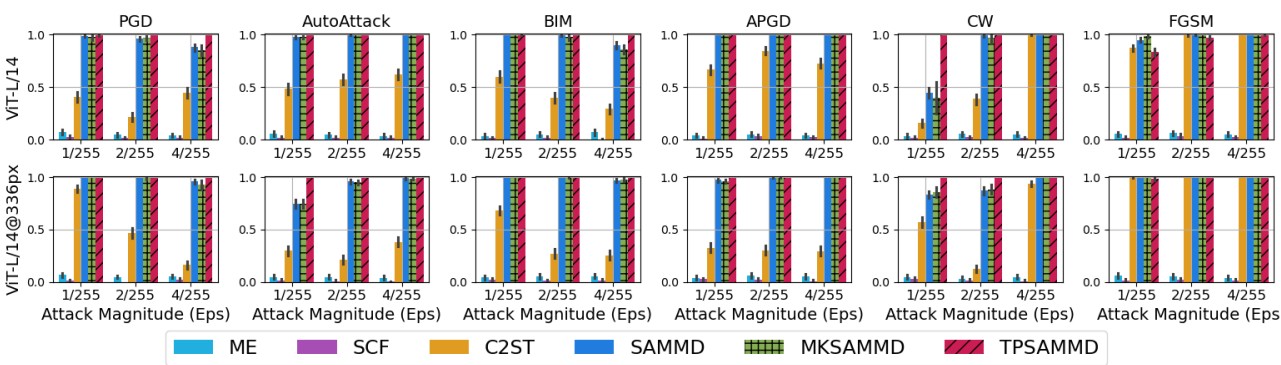

*Figure 20.* Test power of different methods in detecting CIFAR10's adversaries.

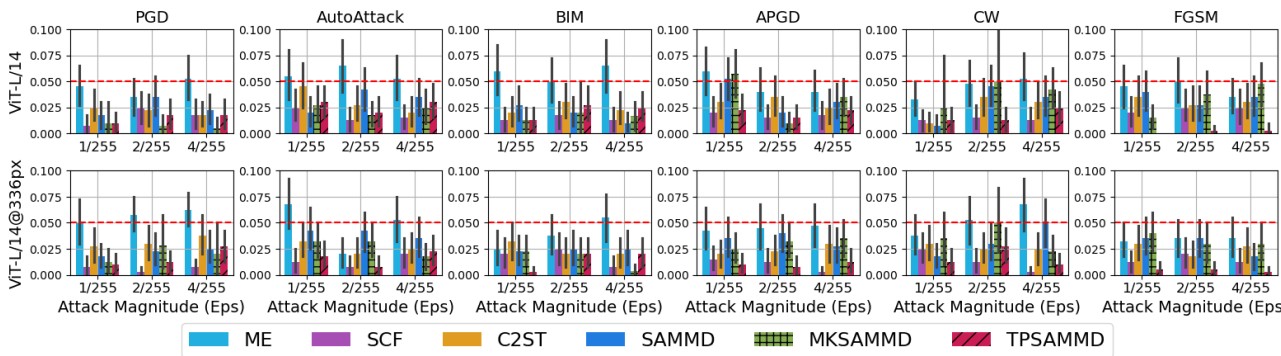

*Figure 21.* Type-I error of different methods in detecting CIFAR10's adversaries.

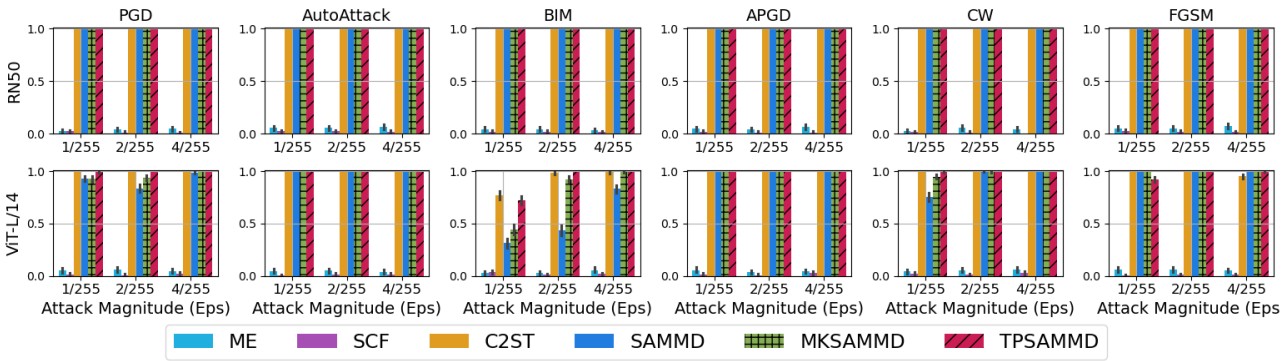

*Figure 22.* Test power of different methods in detecting CIFAR100's adversaries.

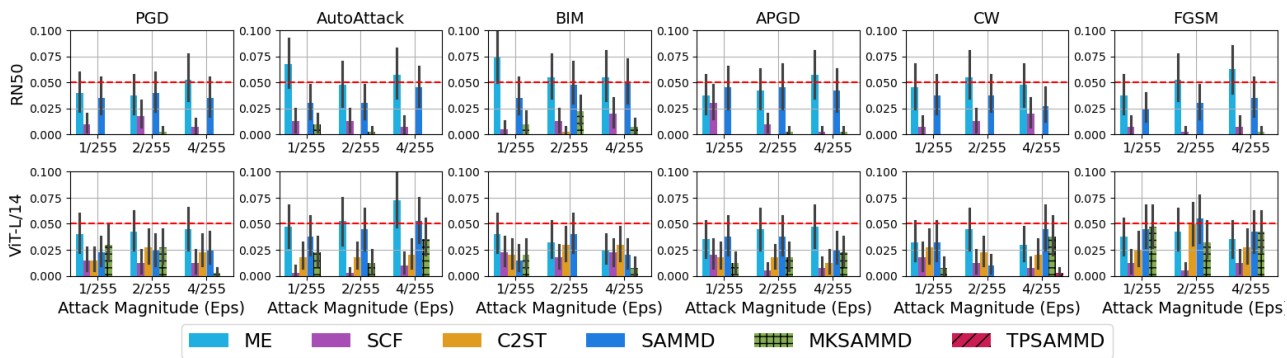

*Figure 23.* Type-I error of different methods in detecting CIFAR100's adversaries.

