# OpenReview forum: "Topological Signatures of Adversaries in Multimodal Alignments"
_ICML.cc/2025/Conference — ICML 2025 poster_

### Official Review · Reviewer_n8xB · 2025-03-11

**Overall Recommendation:** 3

**Summary:**

This paper proposes to measure the topological properties of multimodal alignment from the perspective of image-based adversarial attacks. It introduces two novel topological contrastive losses, based on total persistence and multi-scale kernel methods, to quantify the topological distortion caused by attacks. The theoretical framework, experimental validation, and simulations are well-founded and robust.

**Claims And Evidence:**

Yes, the claims are well supported by clear and convincing evidence.

**Essential References Not Discussed:**

No.

**Experimental Designs Or Analyses:**

The experimental design covers a wide range of common attacks, includes a diverse set of multimodal models, and examines different attack strengths. I believe it is a well-structured and comprehensive experiment.

**Methods And Evaluation Criteria:**

Yes, the evaluation criteria make sense for the problem.

**Other Comments Or Suggestions:**

No.

**Other Strengths And Weaknesses:**

The paper is well-structured with clear visualizations and clear definitions. I thoroughly enjoyed reading such a well-organized and concise work.

**Questions For Authors:**

Is it possible to also study attacks injected into text? Or both?

**Relation To Broader Scientific Literature:**

This paper establishes a strong connection between modern mathematical concepts and the emerging field of multimodal models. It offers a reasonable and precise approach to defining and quantifying the impact of adversarial examples on the alignment of multimodal embedding spaces.

**Theoretical Claims:**

Yes, the theoretical definitions are correct.

---

> ### Author Rebuttal · Authors · 2025-03-28
>
> We thank the reviewer for their positive feedback.
>
> In response to the reviewer's query about attacks on text modality, we conducted additional experiments. In particular, we studied the _confusion prompts_ (also referred to as _cross-class prompt injection_ attacks) [Refs. 1 and 2 below]. We tracked the changes in **Total Persistence (TP)** and **Multi-scale Kernel (MK)** losses as adversarial text was incrementally injected into the data batch (Adv. ratio) (similer to the experiments shown  in the first two columns of Figure 1 for images)  using three different CLIP configurations (RN50, ViT-B/32 and ViT-L/14) on the ImageNet dataset. These losses were computed between the two modalities (Equations (3) and (5) of the paper). Examples of typical benign and adversarial text are:
>
> - _Benign_: "a photo of an apple" (Prediction: "apple")
>   _Adversarial_: "a photo of an apple that resmbles an aquarium fish" (Prediction: "aquarium fish")
>
> - _Benign_: "This is a castle" (Prediction: "castle")
>   _Adversarial_: "This is a castle that mimics a baby" (Prediction: "baby")
>
> _(Here, the prediction "apple" means the CLIP will point the user to a set of images with label apple in the ImageNet dataset)_
>
> Interestingly, we obtain results consistent with our prior findings in images: **Both losses are monotonically changed as more adversarial text are injected into the data batch**:
>
> | | RN50 |RN50| ViT-B/32 |ViT-B/32| ViT-L/14 |ViT-L/14|
> |-----------|-------|-------|-------|-------|-------|-------|
> |  Adv. ratio  | **TP**    | **MK**    | **TP**    | **MK**    | **TP**    | **MK**    |
> | 0.0     | 1.00 | 1.00 | 1.00 | 1.00 | 1.00 | 1.00 |
> | 0.1     | 1.56 | 1.02 | 1.14 | 1.03 | 1.09 | 1.02 |
> | 0.2     | 1.93 | 1.06 | 1.30 | 1.05 | 1.26 | 1.04 |
> | 0.3      | 2.00 | 1.07 | 1.78 | 1.08 | 1.37 | 1.06 |
> | 0.4     | 2.47 | 1.10 | 2.30 | 1.11 | 1.65 | 1.09 |
> | 0.5      | 3.48 | 1.14 | 3.08 | 1.14 | 2.27 | 1.11 |
> | 0.6      | 3.55 | 1.17 | 3.28 | 1.16 | 2.48 | 1.13 |
> | 0.7      | 5.01 | 1.20 | 3.98 | 1.20 | 3.15 | 1.16 |
> | 0.8     | 5.36 | 1.22 | 5.11 | 1.23 | 3.44 | 1.19 |
> | 0.9      | 7.24 | 1.26 | 5.46 | 1.26 | 3.68 | 1.22 |
> | 1.0      | 7.31 | 1.28 | 5.67 | 1.27 | 3.78 | 1.23 |
>
>
>
>
>
> We chose not to include these results in our initial submission for three main reasons. First, we believe the paper is already rich with the presented analyses. Second, a comprehensive examination of the signatures associated with textual attacks would require more thorough investigations due to the variety of attack methods and text-encoding modules, making it more appropriate for future dedicated research. Third, from a detection perspective, given that adversarial attacks significantly alter the original textual content, topological signatures might not be necessary for effective detection of malicious interventions. If the reviewer believes that including these additional results would strengthen the scientific contribution of the paper, we will gladly incorporate them into our final manuscript.
>
>
> We hope the reviewer finds that the manuscript is thorough, well-rounded, and deserving of a higher evaluation than a weak accept.
>
> ---
>
> **References:**
>
> [1] Maus, Natalie, et al. *"Black box adversarial prompting for foundation models."* arXiv preprint arXiv:2302.04237 (2023).
>
> [2] Xu, Yue, and Wenjie Wang. *"LinkPrompt: Natural and universal adversarial attacks on prompt-based language models."* In *Proceedings of the 2024 Conference of the North American Chapter of the Association for Computational Linguistics: Human Language Technologies (Volume 1: Long Papers)*, 2024.

---

### Official Review · Reviewer_9vo5 · 2025-03-17

**Overall Recommendation:** 2

**Summary:**

This paper aims to detect adversarial attacks against multimodal image encoders like CLIP and BLIP, where attackers introduce adversarial perturbations in the image domain to cause misalignment in the text domain (e.g., misclassification). The hypothesis is that, since the primary goal of adversarial attacks is to change the top logit, they would break the intrinsic structure in the data of the (target) label. As a result, such discrepancies can be used to detect attacks. In this work, the authors explored using two topological signatures: Total Persistence (TP) and Multi-scale Kernel (MK). Empirical experiments show that a detector that uses Maximum Mean Discrepancy (MMD) to compare the TP and MK signatures between image logits and textual embeddings has high detection capability against several adversarial attack methods.

## update after rebuttal

The rebuttal has clarified my question about evaluation setup. However, failing to detect adaptive attack using TP-loss increased my worry about the robustness of the proposed consistency measures. For this reason, I remain on the negative side.

**Claims And Evidence:**

* Hypothesis: adversarial attacks will introduce discrepancies at the topological level, when considering embeddings as a point cloud.
* Evidence: the authors proposed two topological-contrastive losses: TP-loss and MK-loss. Experiments on CIFAR and ImageNet shows strong (monotonic) correlations between the number of adversarial images in datasets of specific labels, and the loss between the image and text topological signatures.

**Essential References Not Discussed:**

Did not find.

**Experimental Designs Or Analyses:**

As mentioned above, there are two main complains about the evaluation.

1. While many figures like Figure 3 suggest a monotonic correlation between the proportion of adversarial inputs in the sample set and the proposed losses, they also suggest when the proportion is low, the losses could be small too. However, for practical deployment, it's unclear (1) if there's a principle way to decide the size of the holdout set, (2) the minimum sample size of adversarial inputs for reliable detection, and (3) if there's any tradeoff between the sizes of these two data.

2. Leveraging intrinsic consistency to detect adversarial attacks is a good idea. However, a critical question is whether the proposed consistency measures (i.e., TP-loss and MK-loss) are truly inherit or superficial. Without theoretical support, we usually resort to using adaptive attacks to empirically answer this question. Here, we assume attackers know the detection method and aim to minimize the loss(es) while causing misclassification. If such attacks are not hard, then the measures are likely superficial. On the other hand, if the attacks are hard to construct (within the perturbation budgets), then the features are good.

In addition, several figures suggests the losses are not entirely monotonic for simple FGSM attack, why?

**Methods And Evaluation Criteria:**

* Using intrinsic structure with a distribution to detect adversarial attacks makes sense and have been used before
* While there is no theoretical explanations for the topological signatures, the extensive experiments did show correlation between adversarial samples and the discrepancies between image and text embeddings
* Evaluating setup: 50 attack images and 1000 holdout for CIFAR, 100 attack images and 3000 holdout is fine, and is a more challenging setup than previous work
* Evaluation I wish are included: (1) minimum ratio of adversarial inputs for a reliable detection; (2) adaptive attacks that try to minimize the topological discrepancies.

**Other Comments Or Suggestions:**

Please try to explain the result on FGSM.

**Other Strengths And Weaknesses:**

I appreciate the efforts to explore new mathematical tools to measure consistency of data. But I would want to see, at least some discussions on adaptive attacks, especially the feasibility of optimization-based attacks against the proposed losses.

**Questions For Authors:**

How difficult it would be to launch adaptive attacks to bypass the proposed detection method?

**Relation To Broader Scientific Literature:**

This work explores applying new mathematical tools (topological data analysis) for adversarial attacks detection, which could be an interesting direction to further explore.

**Theoretical Claims:**

N/A

---

> ### Author Rebuttal · Authors · 2025-03-29
>
> We thank the reviewer for the careful examination and thoughtful feedback. We address the following 3 reviewer's concerns:
>
> ## Q1: Number of adversarial and the size of the hold-out data
>
> We agree they are critical factors:
>
> 1. **Number of adversarial samples**:
>    Detection performance depends more on the absolute **number** of adversarial samples ($|Y|$ in Eq. 6) than on their ratio. Increasing the hold-out dataset size ($|Z|$) (effectively decreases the ratio) improves detection by better modeling the logit space.
>
> 2. **Hold-out dataset size**:
>    The size of the hold-out dataset should be sufficiently large to accurately represent the topological structure of the logits. Typically, this size is determined by the dimensionality of the logits—10 for CIFAR-10 and 1000 for ImageNet. We find $|Z| = 1000$ is sufficient for CIFAR-10 (100 points per label). For ImageNet, we chose $|Z| = 3000$ to ensure at least 3 samples per class. However, as demonstrated below, even $|Z| = 1000$ (1 image per class) can be sufficient.
>
> We provide additional experiments using CLIP RN50 with FGSM and PGD to illustrate the above claims:
> https://docs.google.com/drawings/d/e/2PACX-1vTtwOJT6miiHXree45vV0plXMhUj3z5kwflHhQMSXZWT-kd1ssTaYofZ6OgYrmu9jL9waulVseAtQT9/pub?w=960&h=720
>
> As reported in our paper, test power reaches near 1 at $|Y| = 100$ with $|Z| = 1000$. Furthermore, while increasing $Z$ helps for lower $|Y|$ (40–60), the gain diminishes as $|Y|$ grows.
>
> ## Q2: Adaptive attacks
>
> Since **Reviewer M3Mh (Q1)** raised a similar interests, we provided a detailed response in that thread. We summarize the experimental results of an attacker with full gradient knowledge in the following. The attacker enhances FGSM with topological gradients. Specifically, we integrate standard FGSM gradients ($\nabla_{\text{FGSM}}$) and $\nabla_{\text{TC}}$ as follows:
>
> $x_{\text{adv}} = x + \epsilon \left[ (1 - \alpha)\ \text{sign}(\nabla_{\text{FGSM}}) - \alpha\ \text{sign}(\nabla_{\text{TC}}) \right]$
>
> Results with $\epsilon = 4/255$ on CLIP ViT-B/16 in ImageNet:
>
> | α  | Accuracy ↓ | Attack Eff. | Test Power |
> |---|---|---|----|
> | 0.0 | 42.63%     | 1.00        | 0.60       |
> | 0.2 | 42.06%     | 0.98        | 0.27       |
> | 0.4 | 41.00%     | 0.96        | 0.13       |
> | 0.6 | 38.44%     | 0.90        | 0.12       |
> | 0.8 | 31.05%     | 0.72        | 0.05       |
>
> The results indicate that possessing complete gradient information ($\nabla_{\text{TC}}$) allows attackers to circumvent our detection. Nevertheless, realizing the above attack is generally not feasible in practice due to:
>
> - **Unknown $Z$**: Attackers can't reliably match the detector’s randomly chosen hold-out set.
> - **High Cost**: Gradients from Eq. (6) are expensive to compute, especially for iterative attacks.
> - **Loss Specificity**: Evasion for one detector (e.g., Total Persistence) may fail against others (e.g., Multi-scale Kernel).
>
> ## Q3: Topological signatures of FGSM
>
> We agree that FGSM exhibits distinctive behaviors compared to more complex attacks. We elaborate this discussion with the following assumption and observation concerning FGSM:
>
> - **Assumption (A1)**: FGSM, being a relatively simple, generates adversarial logits that less closely resemble the logits of the target class.
>
> - **Observation (O1)**: Although the Total Persistence (TP) measure isn't always strictly monotonic, FGSM typically exhibits an initial steady increase followed by a slight decrease (Figures 10, 11, 13, and most cases in 12).
>
> We argue that **(A1)** explains (**O1**), which is different from the monotonic of more complex attacks. Formally,  (**O1**) means the TP of a adversarial-only point cloud is smaller than TP of mix point cloud in FGSM, i.e., $TP(adv) < TP(mix)$. On the other hand, $TP(adv) > TP(mix)$ for more complex attacks. Our explanation is:
>
> - Assuming we have $K$ labels, the original clean logits point cloud has $K$ clusters. In more complex attacks, the adversarial logits are also that $K$ clusters but have slightly different structures. As the adversarial ratio increases, the $K$-cluster clean point cloud gradually shifts to the $K$-cluster adversarial point cloud. Thus, the TP gradually changes monotonically from $TP(clean)$ to $TP(adv)$.
> - However, due to **(A1)**, we can think of the mixture point cloud of FGSM has $2K$ clusters ($K$ for clean and $K$ for adversarial). As the adversarial ratio increases, the $K$-cluster clean point cloud first becomes a $2K$-cluster point cloud (which can have significantly high TP), then becomes the $K$-cluster adversarial point cloud. This explains why $TP(adv) < TP(mix)$ for some intermediate mixtures. **Nevertheless, we expect this distinctive behavior of FGSM can be better detected by higher-order homology**.
>
> ---
>
> We will include the above discussion in our final manuscript if the reviewer finds them beneficial.

---

> > ### Comment · Reviewer_9vo5 · 2025-04-07
> >
> > Thank you for the additional experiments. Given that the small experiment indeed shows the proposed method may be vulnerable to adaptive attacks, I would not change my score. While what hypothesized in the rebuttal may be true, there are no evidences supporting these claims.
> >
> > > * Unknown: Attackers can't reliably match the detector’s randomly chosen hold-out set.
> >
> > Previous approaches that tried to hide information (e.g., very early days, hiding gradient) has been shown to be vulnerable to surrogate attacks. This may beyond the scope of this work but I personally don't like another attacking paper showing a defense can be bypassed.
> >
> > > * High Cost: Gradients from Eq. (6) are expensive to compute, especially for iterative attacks.
> >
> > There's no quantitative data regarding the computation expense, especially if simple FGSM can succeed, why do attackers need iterative methods?
> >
> > > * Loss Specificity: Evasion for one detector (e.g., Total Persistence) may fail against others (e.g., Multi-scale Kernel).
> >
> > Maybe, but where is the evidence?
> >
> > In general, I appreciate exploring new ways to extract and measure inherit consistencies. Even though the results may be negative, i.e., TP and MK are actually superficial that can be bypassed through adaptive attack, I would encourage the authors to keep exploring.

---

> > > ### Author Response · Authors · 2025-04-07
> > >
> > > First, we disagree with the judgment that "TP and MK are actually superficial." Detection based on signatures is not the sole contribution of our work. The primary contribution is demonstrating the topological signatures of adversarial examples (Sect.3), while the detection component serves as a practical illustration of how our theoretical findings can be applied.
> > >
> > > Second, regarding the question, "If a simple FGSM attack succeeds, why would attackers need iterative methods?": FGSM is generally easier to detect using existing methods (Gao, 2021) that do not specifically target our proposed loss function. Unfortunately, due to the late timing of this question (received only one day before the deadline), we were unable to perform additional experiments to demonstrate this explicitly.
> > >
> > > Third, we believe that the adaptive attack scenario is only peripherally related to the scope of our work. Nevertheless, we have already addressed all previously raised concerns thoroughly.
> > >
> > > Given these clarifications, we respectfully ask for reconsideration of the evaluation score.

---

### Official Review · Reviewer_rmZa · 2025-03-18

**Overall Recommendation:** 4

**Summary:**

The authors consider adversarial attacks that target multi-modal systems relying on alignment of embeddings (the embeddings produced on text and image inputs are supposed to be close to each other, and the attacker supplies, e.g., an image whose embedding is close to an embedding of a wrong, non-matching  text description). They introduce two losses based on Topological Data Analysis tools and show experimentally that these losses are consistently higher for adversarial examples. A mathematical model that can explain this phenomenon is also provided (Poisson Cluster Process). These losses are applied in the Statistical Adversarial Detection (SAD) scenario: given is a set of samples and one must determine whether it contains i.i.d. samples from the same distribution as the training dataset or not (adversarial). The authors suggest to add the gradients of one of their topological losses as an additional feature to the State-of-the-art method for SAD problem (here they exploit the key fact that one can differentiate through persistent homology and propose an algorithm that allows them to efficiently simulate statistical assumptions that are needed and to avoid computing prohibitively ). They demonstrate experimentally that adding these topological features helps increase the performance, especially when the fraction of the adversarial examples in the input is small.

## update after rebuttal

I am going to keep my original score, admitting that I am not an expert in the topic of adversarial attacks and this is probably the reason why the question of adaptive attacks is not a concern to me (my very naive understanding is that, if an adversary is given enough information, it is not surprising that an attacker can circumvent the defense proposed by the authors --- but at a considerably higher cost, I think). I don't have any other concerns with the paper.

**Claims And Evidence:**

Yes.

**Essential References Not Discussed:**

none

**Experimental Designs Or Analyses:**

The description of the experiments sounds pretty reasonable, I don't see any issues here.

**Methods And Evaluation Criteria:**

Yes.

**Other Comments Or Suggestions:**

- Mention explicitly that MK loss is quadratic in the cardinality of persistence diagrams, while Total Persistence loss is linear (there can be many points near the diagonal even for medium-sized inputs, so I would consider this factor when choosing between the two)
-  Calling TP and MK 'homologies' (line 200) is non-standard and confusing, it's better to refer to them as a topological summary. They are both computed from the same homological information.
- 212: 'summation of the difference at all homology groups' -> 'sum of the difference over all dimensions (or over persistence diagrams in all dimensions)'
- 207 'the those' -> 'those'

**Other Strengths And Weaknesses:**

Strengths: the paper is well-written and clearly explains all essential details of the authors' contributions. I very much like the idea of using the gradients of a topological loss as features; I must admit that I have never seen this trick and it is very fitting: taking the derivative can be some sort of an amplifier. The authors discovered a situation in which TDA methods have a good chance to succeed and quite convincingly demonstrated that this is indeed the case with experimental evidence.
Weaknesses: Theoretical explanation (Poisson Cluster Process) is, as the authors say themselves, just a hypothesis confirmed with simulations, not a theorem. This hypothesis only applies to the total persistence loss and does not say anything about MK. Actually, I wonder what are the results of the same simulation for MK loss? Sometimes in the experiments the classifier 2-sample test outperforms both the SAMMD with and its enhanced versions.

**Questions For Authors:**

- MMD is defined for RKHS, so the multi-scale kernel loss fits into it naturally (you just add the Hilbert space into which the diagrams are embedded as a direct summand). Is there similar natural interpretation for total persistence loss?
- It looks like the experimental results in the paper are based solely on dimension 0, e.g., MST. Do the authors expect to gain more from higher dimensions, if computational resources allow that?
- Could the authors even conjecture what causes the monotonic behaviour to flip in Fig.3, row 4? I get that it is still monotonic, but the consistency of the change is intriguing.
- Can one imagine a more sophisticated adversarial attack that targets not just the highest logit, but the whole input? It would be interesting to see whether this can be successfully implemented.

**Relation To Broader Scientific Literature:**

The paper extends a previous method of detecting adversaries by integrating topological features.
From the TDA side, in my opinion, the authors did a very good job by finding a scenario where TDA approach has good chances of making a difference; there is a number of works about applications of TDA to ML and, as far as I know, none of them mentioned this particular case.

**Theoretical Claims:**

N/A (there are no theoretical proofs, only empirical evidence)

---

> ### Author Rebuttal · Authors · 2025-03-30
>
> We thank the reviewer for their positive feedback and insightful comments. We fully agree with the reviewer’s suggestions, including:
> 1. Explicitly mentioning that the Multi-scale Kernel (MK) loss is quadratic,
> 2. Replacing the term *"homologies"* (line 200) with *"topological summary"* to avoid confusion,
> 3. Addressing the identified typos and grammatical errors.
>
> We now address the specific questions raised by the reviewer:
>
> ---
>
> ### Q1: PCP analysis on Multi-scale kernel (MK) Loss
>
> We performed a PCP analysis on the MK loss, following the same setting used for Total Persistence (TP) in Figure 5. The results can be found here:  https://drive.google.com/file/d/1Es536XanO-4MTSgSzimci3NoidjvmzkC/view?usp=share_link
>
> From the results, we observe that the MK loss landscape (analogous to the TP landscape in Figure 5) exhibits less monotonic behavior. While the MK loss generally decreases with increasing $\alpha_s$, it **first increases and then decreases** as the bias ratio $r$ increases. We hypothesize that this non-monotonicity may account for the observed flips in MK loss under certain experimental settings.
>
> Although MK loss is less monotonic than TP, it may serve as a more effective topological summary for adversarial detection in settings where the **spatial distribution** of persistence points carries important information—something MK captures and TP does not.
>
> ### Q2: Hypothesis for the flip in MK loss (Figure 3, Row 4)
>
> As discussed in **Q1**, the MK loss can exhibit both increasing and decreasing trends with respect to the bias ratio $r$, in contrast to Total Persistence (TP), which typically decreases more consistently (see Figure 5).
>
> This suggests that when clean logits shift toward adversarial logits—corresponding to decreasing $\alpha_s$ and decreasing $r$ in the PCP model—the MK loss may behave non-monotonically.
>
> We hypothesize that the decrease (or flip) in MK loss observed in Figure 3 (row 4, for CLIP under AutoAttack) arises because the clean and adversarial logits fall within a region of the MK landscape where the loss decreases as the shift occurs from clean to adversarial. This implies that the behavior of MK loss is more sensitive to the geometric positioning of the logits within the manifold, which is inherently influenced by the dataset and the specific attack method.
>
> ### Q3: Use of higher-dimensional topological features
>
> Through our investigations, we hypothesize that higher-degree homology (e.g., loops or cavities) offers less utility in adversarial detection under the zero-shot setting.
>
> We find that lower-degree homology, especially degree-0, captures the most discriminative structure between clean and adversarial configurations. For example, in our PCP analysis (Figure 4, right side), when the filtration radius reaches half the inter-point distance in a simplex, a degree-1 hole (loop) emerges uniformly across all three configurations. This uniformity reduces its discriminative power. Thus, we believe that focusing on degree-0 information is more effective in our examined tasks.
>
> However, we also observe some unique situations where higher-order topological information helps. That is the situation with FGSM (detailed discussion is in Q3 of Reviewer 9vo5). We argue that, due to the simplicity of the attack, the resulting logits point clouds introduce new adversarial clusters (rather than being near existing clusters of the clean data). As the occurrence of those clusters can be better detected by higher-order summaries, we expect they can help detect that situation.
>
> ### Q4: Attacking methods targeting the entire input
>
> We agree this is a very interesting idea. Given recent advances in TDA, there are promising tools for constructing topological features from raw input data. However, at this time, we are not aware of a straightforward way to design such an attack.
>
> ### Q5: Natural interpretation of TP Loss analogous to MK in MMD
>
> At present, we do not have a natural interpretation of the Total Persistence (TP) loss similar to how MK loss aligns with Maximum Mean Discrepancy (MMD), but we suspect that such an interpretation actually does not exist. The reason is that the multi-scale kernel was originally created (Reininghaus, 2014) to solve exactly this problem of Wasserstein distance (and Total Persistence as a consequence) not embedding naturally into a Hilbert space.

---

### Official Review · Reviewer_M3Mh · 2025-03-18

**Overall Recommendation:** 3

**Summary:**

The paper explores the vulnerability of multimodal machine learning models (such as CLIP and BLIP) to adversarial attacks. It introduces novel Topological-Contrastive (TC) losses—Total Persistence (TP) loss and Multi-scale Kernel (MK) loss—to analyze how adversarial attacks affect image-text alignment. Through the use of persistent homology, the study shows that adversarial perturbations introduce distinctive topological signatures, which can be leveraged to improve adversarial detection. The authors further integrate these topological features into Maximum Mean Discrepancy (MMD) tests, demonstrating that topological-aware MMD tests (TPSAMMD and MKSAMMD) outperform state-of-the-art adversarial detection techniques.

**Claims And Evidence:**

1.	Adversarial attacks alter topological properties of multimodal embeddings – Supported by empirical evidence across multiple datasets (CIFAR-10, CIFAR-100, and ImageNet) and attack types (FGSM, PGD, AutoAttack, BIM, etc.).
	2.	The proposed TP and MK losses capture these topological distortions – Demonstrated through consistent monotonic changes in TC losses as adversarial perturbations increase.
	3.	Integration of topological signatures into MMD-based adversarial detection improves test power – Validated through experiments where TPSAMMD and MKSAMMD outperform standard MMD-based methods.

**Essential References Not Discussed:**

The paper cites most relevant prior work but could expand discussion on the use of topological methods in adversarial defenses outside of multimodal learning (e.g., TDA in adversarial defenses for unimodal settings).

**Experimental Designs Or Analyses:**

•	The experiments are rigorous, using 10,000 samples from multiple datasets and evaluating multiple attack methods.
	•	Results consistently show that topological losses increase with adversarial proportion, reinforcing the main hypothesis.
	•	The MMD-based adversarial detection experiments convincingly demonstrate the utility of the proposed approach.
	•	Ablation studies on different attack magnitudes, architectures, and dataset variations provide further support.

The methodology is sound, and results are statistically significant, with Type-I errors controlled at 5%.

**Methods And Evaluation Criteria:**

The methodology is well-grounded:
	•	Persistent homology is used to extract topological features from multimodal embeddings.
	•	New TC losses (TP and MK) are formulated and tested across different datasets and model architectures.
	•	A novel MMD-based detection approach is proposed, leveraging topological signatures.
	•	Experiments cover multiple attack strategies, model architectures, and datasets, ensuring broad generalizability.

Evaluation is primarily based on:
	•	Monotonic trends in TC losses (validating adversarial impact on topology).
	•	Test power of adversarial detection methods (demonstrating the effectiveness of the proposed methods).
	•	Comparison against strong baselines (existing MMD methods, mean embedding tests, and classifier-based detection).

Overall, the evaluation is comprehensive and well-designed.

**Other Comments Or Suggestions:**

•	Consider evaluating efficiency trade-offs between TP and MK losses for real-world deployment.
	•	Discuss whether adaptive adversaries (specifically attacking the topological loss) could bypass detection.
	•	Clarify computational complexity of topological computations in the main text.

**Other Strengths And Weaknesses:**

Strengths:
	•	Novel application of topological methods to multimodal adversarial detection.
	•	Strong theoretical grounding, supported by empirical and simulation-based validation.
	•	Comprehensive experimental evaluation across datasets, models, and attack types.
	•	Improves over state-of-the-art MMD-based detection methods.

Weaknesses:
	•	Some parts of the theoretical analysis (e.g., Poisson Cluster Process modeling) could be more formally proven.
	•	The computational cost of constructing Vietoris–Rips filtrations might limit real-time applications.
	•	More discussion on potential failure cases (e.g., how robust are the topological losses against adaptive adversarial attacks?).

**Questions For Authors:**

1.	Could adversarial attacks be specifically optimized to fool topological signatures (e.g., adversarial training against TC losses)?
	•	If so, how would the proposed method hold up?
	2.	How computationally expensive are the TC losses compared to standard MMD methods?
	•	Can the method be used in real-time applications?
	3.	Would integrating higher-dimensional topological features improve detection further?
	•	The study primarily focuses on 0-dimensional and 1-dimensional persistence—would considering higher-order homologies be beneficial?
	4.	How does the detection method perform against attacks specifically designed to preserve topological structures?
	•	Could a different type of adversary circumvent the proposed approach?

**Relation To Broader Scientific Literature:**

•	Multimodal adversarial robustness (e.g., adversarial attacks on CLIP/BLIP models).
	•	Topological Data Analysis (TDA) (e.g., persistent homology in machine learning).
	•	MMD-based adversarial detection (e.g., previous work on MMD tests for adversarial detection).

It provides a novel bridge between TDA and adversarial detection, which is original and impactful.

**Theoretical Claims:**

•	The paper provides a mathematical foundation for the TP loss, drawing connections to Poisson Cluster Processes (PCP).
	•	The Wasserstein stability of MK loss is discussed, ensuring its robustness.
	•	The explanation for adversarial perturbations leading to increased TP values is well-reasoned, supported by Monte Carlo simulations.

While the theoretical foundations are strong, a formal proof of why adversarial samples increase total persistence across all settings would further strengthen the paper.

---

> ### Author Rebuttal · Authors · 2025-03-28
>
> We thank the reviewer for careful feedbacks. We address the concerns of the reviewer via the 3 following topics:
>
> ## Q1: Adaptive attack against topological-based detection
>
> Due to the broad landscape of adaptive attacks, we focus this discussion on _gradient-based attacks_ against: an attacker requires the gradient with respect to the topological signatures utilized by our detector (Equation 6). We first discuss why obtaining these gradients is challenging:
>
> 1. **Lack of knowledge of the hold-out $Z$:**
>    An attacker does not typically know the specific hold-out dataset $Z$ used by the detector. Each time the detector runs, it can select a different $Z$ detection. Thus, gradients used by the attacker using a different hold-out set can be significantly different.
>
> 2. **Computational cost:**
>    Even with complete knowledge of $Z$, accurately computing the gradients from Equation (6) is computationally expensive for attacker—especially for iterative attacks. This complexity is from the construction of the Vietoris–Rips (VR) complex. While the detector computes this once, iterative attacks require repeated computations. Note that the attacker also need to back-propagate that gradients on the logit to the input for adversarial perturbations.
>
> 3. **Loss-specific gradients:**
>    The attacker’s gradients must be specific to the loss used by the detector. Thus, gradients for a Total Persistence-based detector may not necessarily enable the attacker to bypass another detection, such as a Multi-scale Kernel detector.
>
> ### Impact of complete knowledge:
>
> Nevertheless, we agree that it is important to study an attacker having complete gradient. Due to the computational impracticality of iterative attacks (point 2), we simulate the attacker using FGSM. We enhance the standard FGSM gradient ($\nabla_{\text{FGSM}}$) with our topological loss gradient ($\nabla_{\text{TC}}$ back-propagate from Equation (6)):
>
> $x_{\text{adv}} = x + \epsilon \left[ (1 - \alpha)\text{sign}(\nabla_{\text{FGSM}}) - \alpha\text{sign}(\nabla_{\text{TC}}) \right]$
>
> Here, α is a parameter balancing the standard and topological gradients. Results with $\epsilon = 4/255$ on CLIP ViT-16/B in ImageNet are:
>
> | α | Accuracy Reduction | Attack Effectiveness | TPSAMMD Test Power |
> |---|---|---|---|
> | 0.0   | 42.63%  | 1.00   | 0.60   |
> | 0.2    | 42.06%   | 0.98   | 0.27  |
> | 0.4    | 41.00%     | 0.96  | 0.13   |
> | 0.6    | 38.44%    | 0.90   | 0.12   |
> | 0.8    | 31.05%   | 0.72    | 0.05   |
>
> As observed, having complete gradient helps the attacker in bypassing our detection, though it comes with a reduction in attack effectiveness.
>
> ### Recommendations for robust detection:
>
> - Utilizing a larger hold-out dataset.
> - Combining multiple detection methods simultaneously.
>
> ## Q2: The usage of higher-dimensional topological features
>
> During our research, we form a hypothesis explaining why higher-degree summary is less effective for adversarial detection. Specifically, we observe that critical distinctions between adversarial and non-adversarial configurations primarily arise from lower-degree homology, particularly degree-0, rather than higher-degree topological features such as loops (degree-1) or cavities (degree-2).
>
> This limitation of degree-1 homology can be shown via our PCP modeling. For example, as shown by the 3 configurations on the right side of Figure 4, when the filtration radius approaches half the distance between vertices of a simplex, a prominent one-dimensional hole emerges across all cases. As this degree-1 feature appears uniformly, it provides limited discriminative power.
>
> However, we also observe some unique situations where higher-order topological information helps. That is the situation with FGSM (detailed discussion is in Q3 of Reviewer 9vo5). We argue that, due to the simplicity of the attack, the resulting logits introduce new clusters (rather than being near existing clusters of the clean data). As the occurrence of those clusters can be better detected by higher-order summaries, we expect they can help detect that situation.
>
> ## Q3: Complexity and real-time application
>
> Compared to benchmark (Gao, 2021), our approach introduces additional complexity due to the integration of topological data analysis (TDA). However, our Maximum Mean Discrepancy (MMD) detection learning (Line 376) operates on logis rather than input data, significantly reducing complexity due to the lower dimensionality of logits. Specifically, in CIFAR10 (about 1,000 points), the additional computational cost from TDA involves 2 VR complex computations, taking around 0.5 seconds on our hardware. In comparison, the training time for MMD is approximately 10–20 seconds. At inference, the combined time for TDA computation and MMD is about 0.5 seconds per sample. For ImageNet (about  3,000 points due to larger hold-out requirements), the additional time introduced by TDA during training is about 10 seconds, which is still lower than MMD training.

---

### Official Review · Reviewer_H6gm · 2025-03-20

**Overall Recommendation:** 4

**Summary:**

In this paper, the authors propose Topological Loss Functions for adversarial detection in multimodal data. The papers begins by discussing the required background. In Section 3, the authors present preliminary evidence that topological features are distinguishing to detect adversaries for multimodal data. The experiments show a monotonic trend of total persistence and multi scale kernel with respect to the adversarial ratio. In Section 4, the authors propose an adversarial detection mechanism based on these topological loss functions and in section 5, they present the results of the proposed detection mechanism.

**Claims And Evidence:**

The claims made in the paper are sufficiently supported.

**Essential References Not Discussed:**

I don't think so.

**Experimental Designs Or Analyses:**

The choice of experiments seems sound for the problem at hand. The authors perform experiments to support their claims of using topological features for adversarial detection by showing that total persistence and multi scale kernel are monotonic functions of adversarial rations. Then, they suggest a method to use these topological features for adversarial detection and perform experiments comparing against existing adversarial detection methods. These experiments seem to make sense to support the claims.

**Methods And Evaluation Criteria:**

The methods and evaluation criteria make sense for the problem that the authors are tackling.

**Other Comments Or Suggestions:**

Line 104 left column: We conduct extensive experiments in -> We conduct extensive experiments on ?

Line 17 Alg 1 in Appendix D: $\nabla_X$ -> $\nabla_Y$?

**Other Strengths And Weaknesses:**

Strengths:

The paper proposes an interesting perspective about alignment in multimodal data as topological similarity between the embeddings.

Weaknesses:

The overall flow in the paper seems ok. However, the flow within each section can be improved. The paper is packed with all sorts of different analyses to which leads to fragmentation. Instead, some of these analyses can be moved to appendix to improve the flow. For example the PCP analysis can be moved to the appendix which would free up the required space to explain the concepts more and improve the flow. I am not suggesting to move it to the appendix. Some thought in this direction can be considered.

**Questions For Authors:**

I am curious to know how the model would perform if the loss function is altered to the distance between vectorizations of the two persistence diagrams instead of the Wasserstein distance. Do you have any experimental results about that?

What is the last degree of homology that you are using the information about?

Did you try Alpha complex filtrations? They are computationally more feasible than VR for $H_0$ and $H_1$.

**Relation To Broader Scientific Literature:**

I think that this work provides an interesting perspective of embedding alignment in terms of topological similarity of the multimodal data embeddings. The authors support their claims with empirical evidence that alignment of topological structure of the embeddings is a distinguishing factor for multimodal data. This opens up a new research avenue to explore more about adding topological regularization terms to make the models robust to adversarial attacks.

**Theoretical Claims:**

The paper does not make any specific theoretical claims.

---

> ### Author Rebuttal · Authors · 2025-03-28
>
> We thank the reviewer for their positive feedback and careful examination of the manuscript. We agree with the reviewer’s comments regarding grammatical issues and typos, particularly the corrections at **Line 104** and in **Algorithm 1** (indeed, $\nabla_X$ should be corrected to $\nabla_Y$).
>
> We now address the following questions of the reviewer:
>
> #### **Q1:** Vectorization of persistence diagrams
>
> #### **Q2:** Meaning and usage of higher-degree homology
>
> #### **Q3:** Usage of Alpha complex
>
> We appreciate these insightful questions, as they are natural considerations of our work. In particular, our research had considered them and we later found out that (perhaps counter-intuitively) these three suggested approaches were less effective for the adversarial detection task presented in our paper.
>
> ---
> ### **Q1 and Q2:**
> We form the following two hypotheses during our research, which explain why the current vectorization approach and the usage of higher-degree homology is less effective:
>
> - **Hypothesis H1 (Addressing Q1):**
>   The main difference between persistence diagrams (PDs) of clean and adversarially perturbed point clouds predominantly appear in *small* (often called *noisy*) features, those located near the diagonal with short lifespans, rather than in dominant, long-lived topological features. Under **H1**, current vectorization methods (commonly emphasize persistent, large-scale features) will fail to capture subtle yet crucial signatures induced by adversarial.
>
> - **Hypothesis H2 (Addressing Q2):**
>   These critical distinctions mainly arise at low-degree homology groups (especially degree 0), rather than higher-degree homology. Higher-degree features, such as loops (degree 1 homology) or cavities (degree 2 homology), tend not to offer substantial discriminative information for adversarial detection (except for some less common cases as FGSM, which we will discuss later).
>
> #### Supporting evidence for **H1**:
>
> Empirical evidence supporting **H1** can be observed by examining the Total Persistence (TP) with different choices of order parameter $\alpha$ (as defined in Equations (2) and (3)). Specifically, a smaller order (e.g., $\alpha = 1$) emphasizes dominant components of the PD, whereas larger $\alpha$ more evenly balance contributions from both dominant and small components. Our results (first column of Figure 1 and the corresponding Section 5) show that using a larger $\alpha$ leads to a clearer monotonic change in TP, translating into significantly better detection performance. Nevertheless, we refrain from further increasing $\alpha$, as doing so might violate the stability property of Total Persistence, as discussed by Divol (2019).
>
> #### Supporting evidence for **H2**:
>
> Regarding **H2**, the highest-degree homology we tested is degree 1, which captures one-dimensional holes. Figure 4 employing PCP modeling adversarial logits can be used to illustrate the limitation of degree-1 summary compared to degree-0. As shown in the 3 plots on the right of Figure 4, when the filtration radius approaches haft the distance between vertices of the simplex, a dominant 1-dimensional hole appears across all configurations. Thus, this feature contains less discriminative power for differentiating those configurations. Our detection experiments further reinforce this claim, demonstrating that homology of degree 0 contributes more to the detection.
>
> However, we also observe some unique situations where higher-order topological information helps. That is the situation with FGSM (detailed discussion is in Q3 of Reviewer 9vo5). We argue that, due to the simplicity of the attack, the resulting logits point clouds introduce new adversarial clusters (rather than being near existing clusters of the clean data). As the occurrence of those clusters can be better detected by higher-order summaries, we expect they can help detect that situation.
>
> ---
> In conclusion, we find that the combination of Total Persistence using the Wasserstein distance, with an emphasis on small-scale features (appropriately chosen $\alpha$), and low-degree homology (specifically degree 0) constitutes a "sweet spot" for adversarial detection tasks.
>
> ---
> ### **Q3:**
>
> The Alpha complex is unsuitable for our problem because it does not scale well to high-dimensional data. Specifically, the dimension of the logits ($X$ and $Y$ discussed in Section 3.1) equals the number of labels predicted by the model, ranging from 10 for CIFAR10 to 1000 for ImageNet. Constructing the Alpha complex typically relies on Delaunay triangulation, which has a complexity of $O(n^{d/2})$, making it computationally infeasible for our scenario.
>
> In contrast, constructing the Vietoris-Rips (VR) complex only requires an adjacency matrix, independent of the data dimension, which is crucial to scale up the experiment to ImageNet. This key difference is the primary reason we prefer the VR complex over the Alpha complex in our work.

---

> > ### Comment · Reviewer_H6gm · 2025-04-07
> >
> > I thank the authors for their efforts in writing a detailed response. I would like to maintain my score.

---

### Decision · Program_Chairs · 2025-05-01

**Decision:**

Accept (poster)

**Comment:**

The paper proposes novel losses, based on topological data analysis, that are sensitive to adversarial attacks on multimodal models. They suggest ways of including these measures to detecting attacks, including ideas novel in the context of TDA+ML, for example, using gradients of the topological losses as features in existing methods for the statistical adversarial detection problem. The authors present a thorough empirical evaluation of their proposed method, convincingly arguing for its strengths. The reviewers raise a number of issues around technical implementation details (including other vectorization schemes, choice of features, adaptive attacks), which the authors carefully address in their rebuttal, including by providing additional experiments. All but one reviewer recommend "weak accept" or "accept". The reviewer recommending a "weak reject" is unconvinced by the authors' rebuttal and is especially concerned with adaptive attacks, which the authors view as peripheral to their work.